# Evolution of Porcine Virus Isolation: Guidelines for Practical Laboratory Application

**DOI:** 10.3390/microorganisms13122658

**Published:** 2025-11-22

**Authors:** Danila Moiseenko, Roman Chernyshev, Natalya Kamalova, Vera Gavrilova, Alexey Igolkin

**Affiliations:** Federal Centre for Animal Health, 600901 Vladimir, Russia; moiseenko_ds@arriah.ru (D.M.); kamalova@arriah.ru (N.K.); igolkin_as@arriah.ru (A.I.)

**Keywords:** porcine viruses, primary cell culture, continuous cell lines, virus isolation, ASFV, CSFV, PRRSV, PCV, PAM, PK-15

## Abstract

Cell cultures are an essential tool for laboratory diagnosis of porcine viral infections. However, interpreting the results requires considering the species and tissue origin of cell lines as well as the specific virus replication characteristics (cytopathic effect). This guide discusses the development of techniques for the primary isolation of viruses from biological material and provides recommendations for culturing viruses in different cell types. According to the World Organization for Animal Health, laboratory diagnosis should aim to isolate the virus in cell culture. We have studied the evolution of virus isolation methods for various diseases affecting pigs, including African swine fever virus (ASFV), classical swine fever virus (CSFV), porcine reproductive and respiratory syndrome virus (PRRSV), pseudorabies virus (Aujeszky’s disease, PRV), rotaviruses (RV), teschoviruses (PTVs), swine pox virus (SwPV), swine influenza A virus (IAVs), parvovirus (PPV), coronaviruses, circoviruses (PCVs), diseases with vesicular syndrome, and others. During our analysis of the literature and our own experience, we found that the porcine kidney (PK-15) cell line is the most suitable for isolating most viral porcine pathogens. For ASFV and PRRSV, the porcine alveolar macrophages (PAMs) continue to remain the primary model for isolation. These findings can serve as a starting point for virological reference laboratories to select optimal conditions for cultivating, obtaining field isolates, and strain adaptation.

## 1. Introduction

Pig farming is a rapidly developing branch of agriculture. Experts project that global meat demand will surpass 400 million tons by 2025. Pork production will hit record growth rates with demand expected to exceed 148 million tons this year [1].

The rising demand for pork products has driven a shift from extensive production systems (e.g., family-operated farms, small commercial farms) to intensive production models (e.g., fattening sites in large-scale industrial pig farms). Industry restructuring has altered the epidemiological patterns of porcine infectious diseases while increasing the economic burden associated with their diagnosis, prevention, and eradication [2].

Diagnostic testing plays a critical role in the system of anti-epidemic measures. The detection of viral infections is possible in three general ways: virological, serological, and molecular methods [3].

The virological method of virus isolation relies on propagating the virus in susceptible biological models (naturally susceptible or laboratory animals, cell cultures, and chicken embryos) [4]. Primary and continuous cell cultures are usually used for the detection of porcine viruses. A key challenge in virus isolation is result interpretation, as many pathogens fail to induce cytopathic effects (CPE) or instead cause non-specific changes (typically cell detachment or destruction), complicating the isolation of pure virus populations. The advantage of the method is its high sensitivity (when supported by additional identification techniques), which ensures the reliability of a positive diagnostic result. The primary drawbacks include substantial costs, high labor requirements, and prolonged processing duration [5].

The serological method evaluates immune responses by studying antibody–antigen interactions [6]. Key advantages of this test include its capacity to detect chronic disease in persistently infected animals and to assess both individual and population-level serological status and seroprevalence [7]. This method has two main drawbacks: (1) it provides indirect evidence of etiology by detecting host immune responses rather than the pathogen itself; (2) diagnosis is often delayed due to the kinetics of antibody production [8,9].

Molecular methods detect viral genomes through various techniques, including the following: polymerase chain reaction (PCR) and its modifications; isothermal amplification methods (e.g., loop-mediated isothermal amplification [LAMP] and nucleic acid sequence-based amplification [NASBA]); fluorescent in situ hybridization (FISH); and genotyping assays [10]. Polymerase chain reaction is a highly sensitive, specific, and easily reproducible molecular biological method [11]. It is a technique that uses enzymes to repeatedly copy (amplify) a specific DNA region in a test tube (in vitro) [12]. Unfortunately, the potential of the method is limited due to its inability to detect the infectious activity of the virus. The limitations of PCR include the risk of false-negative results due to reaction inhibitors in the sample, as well as a high contamination risk during sample preparation, nucleic acid extraction, or amplification [13]. Moreover, molecular studies, as well as virus isolation, require strict adherence to the protocol for the delivery and storage of biological material at −20 °C. An increase in storage temperature results in the degradation of nucleic acids (especially RNA) and destruction of infectious viral particles.

Genotyping, particularly in the context of viruses, involves identifying and differentiating between various genetic variants of a virus by analyzing differences in their DNA or RNA sequences. Key advantages include outbreak investigation, surveillance of sylvatic endemic zones, discrimination between vaccine and wild-type strains, genome certification, and phylogenetic analysis [14]. The resolution of genotyping increases with NGS (next-generation sequencing). This requires a high content of the viral genome, which often requires the preliminary isolation and reproduction of the virus in biological models (naturally susceptible animals, cell cultures, and chicken embryos) [15].

According to the recommendations of the World Organization for Animal Health (WOAH), laboratory diagnostics should be aimed at virus isolation in cell cultures for African swine fever, classical swine fever, porcine reproductive and respiratory syndrome, swine influenza, teschovirus encephalomyelitis, and transmissible gastroenteritis. Moreover, the isolation and subsequent deposition of viral strains for biological characterization can only be achieved using virological methods in reference laboratories and research institutes. In the absence of reference strains in scientific and production settings, developing targeted vaccines for specific prevention becomes virtually impossible [16,17,18,19,20,21].

This review aims to systematize current knowledge and methodologies for isolating field strains of porcine viruses during cultivation. By analyzing published literature and incorporating the authors’ expertise in primary pathogen isolation, the paper provides consolidated recommendations to serve as a reference for researchers, graduate students, and veterinary diagnostic laboratory personnel.

## 2. Sample Preparation for Virus Isolation

Samples for testing should be collected from organs and tissues demonstrating both viral tropism and characteristic postmortem lesions, as these represent the primary sites of viral localization. Clinical material from live animals should be collected from sites exhibiting the most pronounced changes: oronasal swabs in respiratory syndrome cases, rectal swabs in intestinal syndrome cases, the contents of aphthae, vesicles, and pustules in cases of skin and mucosal inflammation, and blood samples in cases of fever and generalized infection. Table 1 presents optimal recommendations for collecting biological material to facilitate porcine virus isolation [16,17,18,19,20,21,22,23,24,25,26,27,28,29].

For the accuracy of the diagnosis, it is necessary to select various organs from carcasses and biological fluids from live pigs in an amount determined by local regulations. After collecting the samples in sterile containers and tubes, they are frozen (−20 °C) and delivered to the laboratory. If it is not possible to store samples at −20 °C, they must be delivered in a thermo-container with refrigerants. It is not recommended to freeze the swabs; they should be stored in a transport medium at 4 °C. Unfrozen samples must be delivered to the laboratory within 2–3 days. Frozen samples should be delivered within a week.

Prior to processing, pathological specimens or blood clots are placed in homogenization tubes containing grinding balls, secured in the adapter, and homogenized by mechanical shaking. If a homogenizer is not available, the material can be ground in a porcelain mortar with a pestle: prior to processing, the material is washed with FBS (pH 7.4–7.6), blotted dry with filter paper, transferred to a porcelain mortar, and manually homogenized using a pestle until a uniform consistency is achieved. However, this method increases the likelihood of laboratory contamination [21].

Then, sterile saline solution is added to the homogenized material or whole blood until a 10% suspension is obtained.

The suspension is transferred to sterile tubes, vortexed, and clarified by centrifugation (1000× *g*, 10 min). The resulting supernatant is collected and filtered through 0.45 μm sterile filters to remove bacterial contaminants. Antibiotics are added to the filtered material (penicillin 500 U/cm^3^, streptomycin 500 mg/cm^3,^ or gentamicin sulfate 500 μg/cm^3^). Then, the sample is incubated at 2 to 8 ° C for 60 ± 10 min [18].

Serum, vesicular or pustular fluid, and nasopharyngeal swabs do not need any preparation. The liquid is filtered through 0.45 μm sterile filters to remove any mechanical impurities and bacterial cells. If the swab was delivered to the laboratory without a transport medium, then before collecting the material, the smear is kept in a saline solution for 15 min, then centrifuged at 3000 rpm for 3 min, and then the supernatant from the arrangements is sent for testing.

The biological material remaining after sample preparation is placed in sterile tubes and stored at −10 to −30 °C until the test completion. It is recommended to keep a collection of archived biological samples in laboratories.

If virus-containing material is not used for further testing, it should be treated with virucidal disinfectants effective against the relevant pathogens, then autoclaved at 131 °C (2 bar) for 90 min before disposal.

## 3. African Swine Fever Virus

African swine fever (ASF) is a contagious, transboundary disease that causes huge economic losses to agriculture in many countries of the world [30]. The clinical and pathoanatomical signs of ASF were first described by R. Montgomery in 1921 in African domestic pigs. He noted splenomegaly, generalized hemorrhagic lymphadenitis, and hemorrhagic syndrome [31]. In 1957, the ASF genotype I virus was introduced to Portugal, after which it spread to neighboring European countries (Spain and France) [32]. After prolonged eradication of ASF, the epidemic in Europe stopped only in the 1990s. However, in 2007, a virulent genotype II virus was introduced from Africa into Georgia, causing the current largest ASF pandemic in Eurasia and the Caribbean [33]. To date, this disease is the most devastating for pig farming [34].

African swine fever (ASF) is caused by the African swine fever virus (ASFV), a large, enveloped, double-stranded DNA virus. ASFV is the only member of the *Asfarviridae* family and the *Asfivirus* genus, and it is also the only known DNA arbovirus [35]. During replication in porcine macrophages, ASFV demonstrates a natural phenomenon of hemadsorption [36].

The initial biological models for ASFV isolating relied on the “hemadsorption” phenomenon. This led to the use of cultures of mononuclear macrophage cells, specifically blood leukocytes and porcine bone marrow (PBM) macrophages [37,38]. The continuous cell lines (PK-2a, PK-15, PK-13, IB-RS-2) and primary porcine kidney cells (PK) required extended ASFV adaptation through serial passages (8 to 73) to achieve high viral loads and consistent cytopathic effects (CPEs) [39,40,41,42,43]. While monolayer cultures of porcine kidney cells, chicken embryo fibroblasts, and baby hamster kidney (BHK-21) cells were useful for studying the morphology and ASFV replication cycle using electron microscopy and plaque assays, they were not suitable for the primary isolation and identification of field isolates [43,44,45,46,47,48]. A comparative analysis of the permittivity of 16 different cell cultures showed the ability of ASFV replication in cells of pigs (alveolar macrophages, swine testicular (ST) cell line), green monkeys (Vero, CV2, COS-1), and rhesus macaques (LLC MK2), and no replication in cell cultures of hamsters (BHK-21, CHO), humans (HeLa, Sh-SY5Y), mice (A 9, 3T6, NP3, L 929), dogs (MDCK), and fall armyworm (Sf9) [49]. Thus, in the second half of the 20th century, it was established that ASFV has a narrow specificity for propagation in specialized cells (macrophages) of the *Suidae* family [50].

A little later, tropism of ASFV to primary cultures of pig endothelial cells was discovered after trypsinization of the thoracic aorta [51,52]. However, obtaining monolayer endothelial cultures is an inefficient and costly process, so it was abandoned in favor of macrophage cultures.

Despite ongoing efforts to identify ASFV-sensitive cell lines, the microtitration technique using primary porcine bone marrow (PBM) cells remained the standard method for virus detection and viral load quantification in biological samples for an extended period [53]. However, beginning in the early 1980s, primary cultures of porcine alveolar macrophages (PAMs) gradually replaced pig bone marrow cells due to their lower cost, easier preparation, and ability to be cultured in roller bottles, which enabled higher ASFV titers compared to a stationary culture system [54]. In addition, alveolar macrophages were successfully used in plaque assays [55]. However, due to frequent fungal contamination and low cell yield, PAMs failed to meet initial expectations. Consequently, researchers developed genetically engineered immortalized porcine cell lines (3D4, IPAM, WSL, and ZMAC-4) from these macrophages that demonstrated permissivity to ASFV field isolates [56,57,58]. The method consists of transfecting primary macrophages with plasmids that contain a large T-antigen of the SV40 virus. This suppresses the activation of anti-oncogenes and apoptosis (p53 and pRb). The malignant transformation of macrophages occurs as a result of immortalization. A study comparing ASFV protein expression in primary pig macrophage cells and the WSL-R cell line showed no significant differences in their sequence, structure, or synthesis levels. This finding is crucial for the initial identification of ASFV isolates and their subsequent storage in collections [59].

In addition to PAMs, tissue macrophage cell cultures have also successfully demonstrated their ability to isolate ASFV. The primary trypsinized porcine splenocyte (PS) culture demonstrated superior performance in comparative studies, achieving high viral titers (>6.5 lg HAD_50_/mL) and the shortest ASFV replication cycle (12–24 h) when contrasted with PBMC (48 h), PK (48 h), and ST cells (48–72 h). These data highlight the diagnostic potential of PS for ASFV [60]. Porcine renal macrophages are also permissive to ASFV. Recently, Masujin K. et al. obtained a porcine renal macrophage cell culture (IPKM) that was permissive to virulent strains of various genotypes [61,62].

Since the total yield of monocytes from porcine peripheral blood mononuclear cells is low, this technique has not been used in the preparation of cell cultures in laboratories specializing in ASFV. However, in Russia, a transferable A_4_C_2_/9k line was obtained by hybridization of SPEV (porcine embryonic kidney line) and porcine primary leukocytes. All field isolates of the ASF virus obtained in Russia from 2008 to 2013 were replicated in these cells [63].

Although existing continuous cell lines of non-macrophagal origin have shown limited success in viral cultivation, researchers are pursuing novel experimental strategies to bypass the cell-type tropism that constrains viral replication to specific cellular hosts [56]. Hurtado C. et al. found that COS-1 cells, derived from CV-1 cells immortalized with SV40, are susceptible to various virulent strains of ASFV [64]. However, the viral load in COS-1 was lower than in primary porcine macrophages, which affected its local and rare use in laboratories. While the MA-104 cell line derived from African green monkey kidney cells demonstrates permissivity to ASFV field isolates with a Ct value of 22.14, its viral load is significantly lower compared to PAMs [65,66]. In addition, when ASFV adapts to African green monkey kidney cell lines, the titer critically decreases after a series of passages. This requires the so-called “intermittent passage”, where the viral material obtained from a low-permissivity culture (Vero, MA-104, CV-1) is inoculated into a high-permissivity primary cell culture (PBM, PAM, PS), to increase the infectious dose in subsequent passages. This process helps to prevent the loss of titer during virus adaptation.

The potential applications of porcine cell subcultures remain poorly characterized, highlighting the need for enhanced virological methods in ASF research [67]. The porcine synovial membrane (PSM) subculture obtained from the carpal articular surface is not widely used for ASF diagnosis due to low viral load (4.0 lgHAD_50_/mL), which restricts the sensitivity of the method [68].

Thus, ASFV isolation in primary pig cells is still a relevant method. The diagnostic utility of permissive continuous cell lines (WSL-R, COS-1, A_4_C_2_/9k, MA-104, and others) remains controversial and requires further validation. And, since the gold standard for ASF diagnosis according to the WOAH involves isolating the virus in permissive cell cultures and then identifying it using the hemadsorption (HAD) test, the use of this method in reference laboratories is essential [16]. And, despite the widespread use of PCR in routine diagnostics, virus isolation remains an important stage for subsequent scientific research. Thus, the study of the strain’s biological properties becomes possible only after its isolation [69]. In addition, to understand the comprehensive view of ASFV genotypes circulating, it is necessary to conduct whole-genome analysis using second- and third-generation sequencing (NGS or HTS). ASFV isolation and accumulation are necessary for the purpose of preparing samples of genomic DNA and obtaining high genome coverage during assembly [70].

However, it is worth remembering that during the ASFV genotypes I and II circulation in Eurasia, non-hemadsorbing strains (for example, Lv/17/WB/Rie1) were also isolated. This requires the PCR study of culture material obtained in the final passage (3–5) [71]. The lack of confirmation of virus isolation by PCR carries risks associated with false-negative results at diagnosis and “omission” of non-hemabsorbing ASFV variants. In addition, the phenomenon of hemadsorption is characteristic not only of ASFV but also of some others (for example, paramyxoviruses). Therefore, identifying ASFV by PCR after each passage is always preferable. A quick guide to ASFV isolation from biological material is shown in Figure 1.

## 4. Classical Swine Fever Virus

Classical swine fever (CSF) is a highly contagious disease of pigs characterized by hemorrhagic syndrome, fibrinous pneumonia, colitis, and high mortality (70–100%) [72]. The first CSF outbreaks were reported in Ohio (USA) in 1833. In the 1860s, the virus was introduced into Great Britain and then spread throughout Europe, Asia, and Oceania, as well as Latin America [73]. However, in the second half of the 20th century, vaccine strains were developed (Rovac and C, and their derivatives), which have proven to be effective and safe [74]. Nevertheless, despite widespread vaccination against CSF in China, Russia, India, Japan, and Latin America, new subgenotype 2.1 is able to infect pigs vaccinated with live vaccines derived from strain C [75]. In this regard, the epidemic situation continues to be one of the most significant threats to pig farming.

The CSF etiological agent is the enveloped +RNA *Pestivirus suis*, a member of the genus *Pestivirus*, family *Flaviviridae* [76]. Most CSFV isolates and strains do not exhibit CPE during propagation in cell cultures [77,78,79]. This may be attributed to the RNase activity of glycoprotein E^rns^. However, this is just a hypothesis that has not been reliably confirmed subsequently. Notably, deletion of this glycoprotein has been shown to reduce CSFV virulence in domestic pigs [80,81,82].

For a long time, CSFV was isolated and propagated using naturally susceptible animals (domestic pigs) [83,84]. The first attempts to adapt the virus to growth in primary porcine embryo or PK cell culture were unsuccessful [85,86].

In the absence of effective methods for CSFV identification in cell cultures, an indirect END (exaltation of Newcastle disease virus) method was used [87]. For that, ST cells were inoculated with CSFV, and on day 4 post-inoculation, 6.0 lg PFU (plaque-forming units) of Newcastle disease virus were added to the culture medium. The CSFV-positive samples exhibited a characteristic CPE on days 3–4 after combined cultivation, while CSFV-negative samples developed CPE on days 5–6. The method was subjected to multiple modifications; its sensitivity significantly increased, and it is now comparable to that of the porcine bioassay [88,89,90].

The problem of CPE detection can be addressed by employing the dome disappearance method in FS-L3 cells [91]. Following CSFV infection of a confluent monolayer of cells shaped as proximally located large domes, complete cell degeneration and disappearance occur by day 5 post-infection. The accumulation titer is 6.8–6.9 lg TCID_50_/mL [92].

The development of electron microscopy and serological techniques (direct immunofluorescence assay (DIF) later made it possible to detect the CSFV replication and, thereby, determine the permissiveness of cell cultures to CSFV [93,94,95]. Thus, it has been proven that CSFV demonstrates high replication activity in cells of the spleen, bone marrow, blood leukocytes, macrophages, monocytes, and lymphocytes [96,97]. A correlation has also been identified between the permissivity of cell cultures to the CSFV and the activity of non-specific esterases [98].

In the diagnosis of CSF, continuous cell lines of porcine origin (PK-15, SK-6, CPK-NS) or primary cultures of porcine testicular cells (ST) are mainly used [77,78,79,99,100,101,102,103]. The titer is expressed in cell culture infectious dose (CCID) units.

PK-15 cells are widely recognized as the most extensively studied and commonly utilized cell culture for CSFV diagnosis [79,99,100]. However, already in the 1990s, heterogeneity in the permissive properties of different PK-15 cell sublines was noted, probably due to cross-contamination (by cells of other lines or organisms) in laboratories. As a result, cells with the highest sensitivity to CSFV were cloned, such as clone PK(15)A, obtained at the European Reference Laboratory for CSF, and clone PK-15/B5, obtained in Russia. These clones are recommended for use in diagnostic studies for CSF [104,105].

The continuous porcine kidney cell line SK-6 is also widely employed as a cellular substrate for CSFV isolation [106,107,108]. The continuous kidney cell line of newborn piglet (CPK-NS) has also proved to be promising. Notably, specific CPE were observed following infection with both cytopathic and non-cytopathic strains of CSFV. The accumulation titer reached 6.9–7.3 lg TCID_50_/mL and correlated well with the immunoperoxidase technique (IPT) results [78].

The highest level of CSFV replication is achieved in primary trypsinized cell culture ST (up to 7.5 log CCID_50_/mL), which seems to be more optimal for diagnosing the disease in a virology laboratory [79,101,105]. Some other primary cell cultures (PS, PK, PBMC, cell subcultures and diploid cell strains (PSM), swine spleen (SSs), continuous cell lines (pig embryonic kidney (PEK)), hybrid of porcine embryo kidney (SPEV) and porcine lymphocytes (A_4_C_2_), porcine kidney (IB-RS-2), and Siberian ibex kidney (PSGK) of porcine origin, as well as a subculture derived from sheep testicles) are permissive to CSFV, but low viral titers of field isolates (<5.0 lg CCID_50_/mL) may yield unreliable results, making them unsuitable for precise diagnosis [67,68,98,101,109,110,111,112,113,114,115].

Viral isolation with subsequent confirmation by means of various detection assays represents the optimal diagnostic approach for reference laboratories and research institutes. This method is issued by the WOAH as a reference standard and demonstrates superior sensitivity and specificity compared to alternative techniques for CSFV isolation and accumulation, while also enabling the deposition of viral isolates into state collections [116]. However, the routine use of viral isolation assays for CSF diagnosis is limited by several factors: high costs associated with cell culture maintenance, stringent sterile conditions, the need for additional virus identification methods (due to the absence of CPE), and prolonged duration (up to 3 weeks). Furthermore, similar to other pestiviruses, CSFV poses significant contamination risks in continuous cell line cultures due to a lack of cytopathic activity when co-cultured with other viral pathogens [117,118]. Viruses that cause bovine viral diarrhea (BVDVs) are often found in fetal bovine serum and can contaminate cell lines. It is therefore important to conduct intralaboratory monitoring to identify such serum samples and eliminate them promptly. All these circumstances require the competent use of additional methods for CSFV identification, including IF (immunofluorescence assay) and IPM (immunoperoxidase method), as well as PCR. A quick guide to CSFV isolation from biological material is shown in Figure 2.

## 5. Pseudorabies Virus

Aujeszky’s disease (pseudorabies) is a persistent viral infection affecting multiple animal species, primarily pigs. In 1813, a disease causing severe itching was first reported in the United States [119]. In 1902, Aladár Aujeszky presented a scientific justification for the disease, describing the successful isolation of the virus in bulls, dogs, cats, pigs, and rabbits [120,121,122]. Due to the lack of accurate diagnosis methods in 1947, the virus spread to pig farms in China [123]. The disease has zoonotic potential; the first reports of human disease were recorded in 1914 with clinical signs such as fever, weakness, and dysphagia [124]. In 2017, China reported the first case of human endophthalmitis caused by the pseudorabies virus (PRV) [125]. To date, there have been increasing numbers of cases of people infected with this disease in several provinces in China. Interestingly, all of these patients were associated with working with pigs or had injuries while working with affected animals or pork products [124]. Thus, it is necessary to conduct systematic laboratory monitoring of pseudorabies mainly in pigs in order to exclude the infection of pig breeding workers with PRV.

The causative agent is *Varicellovirus suidalpha 1*, an enveloped, double-stranded DNA virus belonging to the genus *Varicellovirus*, subfamily *Alphaherpesvirinae*, family *Orthoherpesviridae.* The infection predominantly targets the respiratory tract and central nervous system of pigs [126]. Unlike many porcine-specific viruses, pseudorabies virus (PRV) exhibits pantropism—characteristic herpesviral features—along with a wide host range including rabbits, rodents (mice, rats, ferrets), equids, bovines, canines, felines, and other homoiothermal animals [127,128,129,130,131,132,133,134,135,136]. High levels of PRV replication have been demonstrated in dolphin kidney cell cultures, suspended duck and quail embryonic fibroblasts, mink lung cells, and many other continuous cell cultures, which indicates high adaptability and highlights a wide range of possibilities of the virological method in disease diagnosis [137,138,139].

In 1946, Buchnev et al. proposed a live vaccine from a virulent strain of the Aujeski disease virus, weakened more than 800 times by passage in a cell culture of chicken embryos, and named it Bucharest [BUK]-dl3 [140]. After that, isolation in primary cultures and continuous cell lines of chicken embryo fibroblasts (CEF) is recognized as the gold standard for PRV isolation. The high permissivity established since the early 1960s enabled the prolonged use of this method for PRV propagation in both adhesion and suspension systems, as well as in the modified plaque assay protocol [138,141,142,143]. In 1961, Adorján Bartha described a low-virulence field strain, Bartha-K61, obtained by serial passage on kidney epithelial cells from calves and pigs, as well as on chicken embryos, marking the beginning of an era in pig vaccination against Aujeszky’s disease [144,145]. Furthermore, several studies demonstrate that chicken embryo fibroblasts offer superior advantages over primary porcine kidney cells for the laboratory diagnosis of Aujeszky’s disease [146]. CEF cultures are prepared using Hanks’ balanced salt solution supplemented with 0.5% lactalbumin hydrolysate and 5–10% serum (equine, bovine, or avian origin), offering a cost-effective alternative [143,147].

Despite domestic pigs being the primary natural reservoir for PRV, porcine epithelial cell cultures do not show increased viral replication capacity, with ST primary cultures reaching a maximum titer of only 4.38 lg TCID_50_/mL [139,146]. While the newborn pig kidney (NPK) cell line demonstrated permissivity to PRV, it never gained diagnostic relevance [148]. Nevertheless, high replicative activity was observed in two continuous porcine cell lines: NSK (newborn swine kidney) and NPTr (newborn piglet tracheal mucosa), achieving final titers of 7.5 lg and 7.74 lg TCID_50_/mL, respectively [149]. The high sensitivity of continuous porcine alveolar macrophages (IPAM) to PRV (accumulation titer of more than 7 lg TCID_50_/mL) was also demonstrated. However, the high cost of the RPMI-1640 medium required for IPAM cell growth and maintenance, as well as the substantial fetal bovine serum content requirement (up to 15%), significantly increases diagnostic expenses. These factors diminish the prospects of this model compared to simpler cultivation systems for diagnosis [57].

At the same time, PRV is not replicated in all cultures. A few attempts to isolate PRV using BT cell culture (bovine turbinate tissue cells), which is quite popular in virological studies, have demonstrated limited success, achieving only low viral titers (~2.63 lg TCID_50_/mL) [139,150]. While not commonly employed for primary isolation, both mouse peritoneal cells (MPC) and bovine kidney cells (MDBK) have demonstrated permissivity to PRV replication [151,152].

Cultures of non-porcine origin can also be successfully used for PRV isolation. Thus, a high cytopathic activity was shown after six passages of PRV adaptation to BHK-21, with titers of infectious activity reaching 6.5–7.25 lg TCID_50_/mL [153]. The most well-characterized Vero cell line shows potential as a substrate for PRV propagation [154,155,156].

In recent decades, progress in the study of PRV genetics and molecular biology has allowed researchers to identify the main virulence factors and prove that the gene *gE* is responsible for neurovirulence in pigs [157,158,159]. Using genetic engineering, Quint et al. successfully deleted the *gE*-coding sequence from the genome of a highly virulent field strain, NIA-3, in the United States and created an attenuated “marker” vaccine strain 2.4N3A [160]. A monolayer continuous baby hamster kidney cell culture (BHK-21) has been recommended as an in vitro model for cultivating vaccine *gE*-negative strains included in marked vaccines [161,162,163,164].

Currently, researchers are working to improve existing vaccines and simplify the immunization process by creating genetically engineered vaccines. These vaccines provide protection and do not cause any clinical signs in pigs [165].

PK-15 is the most studied and recommended cell culture for PRV isolation from biological material [22]. At the same time, depending on the multiplicity of infection (MOI), the total viral load in titers varies from 4.0 lg to 8.5 lg TCID_50_/mL [139,150,166]. A lower priority alternative to PK-15 may be SK-6 [167,168]. CPE can be recorded in 18–72 h post-infection and is characterized by increased optical cell density, symplasts and syncytia formation, complete monolayer detachment, and degradation. Since PRV-like CPE may occur during infection with other viruses, pathogen identification should be confirmed by means of an IPT, indirect immunofluorescence assay (IFA), or virus neutralization test (VNT) [22]. A quick guide to PRV isolation is shown in Figure 3.

## 6. Porcine Rotaviruses

Porcine rotavirus infection is an acute, highly contagious disease affecting newborn piglets that is characterized by vomiting and diarrhea. Porcine rotavirus (RV) was first identified via electron microscopy in 1975; it was derived from the intestinal contents of piglets that died with diarrhea [24]. In subsequent years, infection was reported in humans, cattle, ferrets, and recently dogs [169,170,171,172]. The World Health Organization (WHO) estimates that rotavirus infection causes approximately 450,000 deaths among humans each year, with more than 90% of these deaths occurring in developing countries in Asia and Africa [173]. In addition, porcine rotaviruses have zoonotic potential [174].

The virus has become widespread, causing economic losses due to reduced productivity among piglets. Researchers have proven that 70–85% of the blood sera studied from animals in France and Japan contain antibodies to rotavirus. In all cases, in selected sera from adult pigs using the enzyme-linked immunosorbent assay (ELISA), anti-rotaviral antibodies were detected in Denmark and farms in Venezuela [175,176].

The causative agents are porcine rotaviruses belonging to the genus *Rotavirus* within the family *Sedoreoviridae*. Based on the VP6 gene, groups A, B, C, E, and H are considered pathogenic to pigs [177]. While RVA, RVB, and RVC are ubiquitous, RVE has only been circulating in the United Kingdom, and RVH has circulated in Brazil, the USA, and Japan since at least 2002. The genome consists of 11 segments, resulting in the constant reassortment of rotavirus within the group [174]. Two different VP3 lineages have been identified in humans and the human VP6 lineage in pigs, indicating reassortment between rotaviruses in pigs and humans [178,179]. Genotyping within each group is based on segments of outer capsid proteins VP7 (G-types) and VP4 (P-types), with other segments being used less frequently for genotyping as virus-neutralizing antibodies form specifically against VP7 and VP4 [174].

Porcine rotaviruses replicate intensively in small intestine epithelial cells and, to a lesser extent, in cells of the caecum and colon [180]. Despite its characteristic localization in the small intestine, rotavirus replicates well in pig kidney cell cultures. Therefore, for the first time, a rotavirus was isolated in primary PK cell culture in 1977 [181]. Rotavirus accumulation and passaging exhibited a moderately pronounced cytopathic effect, and therefore, the IFA was applied for the virus infectious activity control [181,182]. It is also demonstrating efficient replication in SPEV cells under roller cultivation, with the viral titers reaching 8.0 lg TCID_50_/mL within 24 h post-inoculation, which indicates the high sensitivity of this method [183]. When the IPEC-J2 cells (porcine small intestinal epithelial cells–jejunum 2) were infected with the OSU rotavirus strain at a multiplicity of infection (MOI) of 20 PFU/cell, the rotavirus titer was 6–7 lg PFU/mL [184].

It was noted that RV field isolates are successfully adapted to grow in a continuous MA-104 cell line with CPE. Within 18 h post-infection, infected cells exhibit granularity, loss of adhesion, and nuclear pyknosis, which indicates profound rotavirus-induced alterations at the cellular level. Since the 1980s, most laboratories have isolated rotaviruses using MA-104 [185]. Rotavirus HN03 strain demonstrated efficient reproductive properties in MA-104 cells, inducing CPE in >80% of the cell monolayer by day 5 post-inoculation, while the virus titers varied from 7.1 lg to 8.1 lg TCID_50_/mL in passages 4–10. The maximum viral titer of 8.1 lg TCID_50_/mL was recorded 27 h post-inoculation [186].

These findings demonstrate that the MA-104, SPEV, IPEC-J2, and PK cell lines are permissive to rotavirus infection and represent suitable systems for RV isolation and in vitro propagation.

Due to the higher viral load, we recommend isolating porcine rotaviruses in the MA-104 cell line. The most optimal option is pre-sample preparation—incubation of a viral suspension with trypsin (10 µg/mL) or another proteolytic enzyme (for example, pancreatin). The enzyme cleaves the capsid protein VP4 into VP5 and VP8, which increases the infectivity of rotavirus [187]. Also, during cultivation, a high concentration of trypsin in the medium (25–30 µg/mL) should be maintained to increase the viral load. A quick guide to porcine rotavirus isolation is shown in Figure 4.

## 7. Swine Influenza Virus

Swine influenza is a highly contagious acute disease that causes epidemics in most regions of the world. It causes high morbidity (up to 100%) and low mortality (10–15%) among infected pigs, resulting in significant economic damage to pig farms due to decreased meat production [188]. Swine influenza control mainly consists of vaccinating sows to protect their piglets with colostral antibodies [189]. In recent decades, controlling swine influenza has become more difficult through vaccination due to virus evolution, which has led to a reduced effectiveness of existing vaccines. Commercial swine influenza vaccines that are effective against homologous strains provide cross-protection, but they have become less effective against evolving viruses.

The causative agent, influenza A virus, belongs to the genus *Alphainfluenzavirus* within the family *Orthomyxoviridae*, and it is an enveloped RNA virus with an eight-segmented genome, where each of these segments is associated with proteins to form a helical nucleocapsid. Virus subtypes are classified according to the hemagglutinin (HA) and neuraminidase (NA). These are the most important glycoproteins of the influenza virus, which affect its virulence. The virus enters target cells via receptor-dependent endocytosis by binding hemagglutinin to sialic acid [190,191]. Antibodies to HA are virus-neutralizing. NA cleaves the bond between sialic acid and HA during virion exocytosis from the target cell.

Influenza was first recognized as a respiratory disease of pigs during the human Spanish flu pandemic in 1918 and was classified as H1N1 [192]. In 1979, swine influenza appeared in Belgium and Germany [193]. This virus has been circulating steadily among pigs and spread to many European countries as well as China and Korea [194]. Seasonal reassortment of avian and human influenza viruses with endemic swine influenza virus H1N1 resulted in multiple subtypes of H3 and H1. It turns out that in 1978, the avian H1N1 virus overcame the interspecies barrier, passing from wild ducks to pigs in Europe and transferring the hemagglutinin segment of lineage 1C [195]. In the late 1990s, a new triplet reassortment H3N2 appeared in North America and became dominant among pig populations [196]. Thus, currently there are three subtypes circulating: H1N1, H1N2, and H3N2.

In 1931, researchers were able to isolate the virus from infected pigs with clinical signs of respiratory tract damage. Selected nasal discharge was used to inoculate ferrets, and in 1937, the virus was isolated in chicken embryonated eggs (CEE) [197]. The high permissivity and viral load of the influenza virus in the CEE model showed a significant advantage over cell cultures. This model limited the study of the permissivity of various continuous lines to the virus because it was not practical. So, in comparison to biological systems for H1N1 virus isolation, such as CEEs and the continuous Madin–Darby canine kidney (MDCK) cell line, CEEs demonstrated significantly higher susceptibility that was confirmed by the McNemar test, where the significant difference between isolation methods was determined at levels *p* ≤ 0.001) [198]. This data highlights CEE’s critical role in laboratory diagnosis of swine influenza. Analysis of viral cultivation time after inoculation into CEEs also revealed some deviations: the maximum viral load occurred 3–5 days post-inoculation, with approximately 75% of samples yielding positive results, whereas by day 7, only 15% of samples retained infectious activity. These findings underscore the importance of optimizing cultivation time, as it critically impacts viral isolation efficiency [198]. The H1N1 influenza virus replicates efficiently in chicken embryonated eggs (CEEs) following inoculation into either the allantoic or amniotic cavity, without causing embryo mortality. CEE-adapted strains induce spot-like lesions on the chorioallantoic membrane (CAM). When CEEs are infected via the allantoic cavity, hemorrhages appear on the body of the embryos. Cultivation is usually performed in 10–11-day-old CEEs. The maximum virus accumulation occurs in 48–72 h post-inoculation, and its titer in allantoic or amniotic fluid usually reaches 10 lg EID_50_/mL (embryonic infectious doses) [199].

Since 1968, MDCK has also been used to isolate swine influenza viruses (IAVs) [200]. In 1975, it was found that adding trypsin (20 µg/mL) to the medium increased the viral load during IAVs replication in MDCK, making it possible to conduct virological research in laboratories that were not equipped to work with CEE [201]. However, because of the reduced sensitivity of MDCK cells to swine viruses compared to CEE, this model is not used for vaccine production.

Studies of viral replication across different cell lines revealed significant variations in both susceptibility and replication. Thus, clones derived from the human colon CaCo-2(2) line exhibited a significantly higher H1N1 virus replication level compared to the CaCo-2(1) clone, highlighting the importance of cell line differentiation when optimizing viral cultivation conditions [202]. The continuous newborn pig trachea (NPTr) cell line showed high permissivity to the swine influenza virus (IAVs), with stable viral replication over 12–48 h and an average titer of 5.03 lg TCID_50_/mL. In contrast, when replicated in PAMs, influenza virus titers peaked at 24 h post-inoculation (hpi), averaging 2.70 lg TCID_50_/mL [203].

Thus, IAVs are capable of replicating in various biological systems, including chicken embryonated eggs, continuous canine kidney cell lines, human colon adenocarcinoma cells, and newborn pig trachea cells. CEEs exhibit a higher susceptibility to the virus compared to cell cultures and remain the preferred biological model for isolating IAVs. If the laboratory does not have the facilities and equipment for working with CEEs, then it is acceptable to use MDCK. The isolation of new IAV genovariants plays a crucial role in the continuous improvement of inactivated vaccines. It is worth noting that work involving IAVs should be performed in BSL-2 laboratories with some elements of BSL-3 (BSL-2 enhanced conditions). A quick guide to swine influenza virus isolation in CEEs is shown in Figure 5.

## 8. Porcine Reproductive and Respiratory Syndrome Virus

Porcine reproductive and respiratory syndrome (PRRS) is a contagious viral disease characterized by reproductive failure in breeding stock (including late-term abortions and stillbirths) and respiratory disease in young pigs.

The causative agent is a single-stranded positive-sense RNA virus of the family *Arteriviridae*, genus *Betaarterivirus*. The pathogen is classified into two major species (autonomous species): European (subgenus *Eurpobartevirus*, species *Betaarterivirus europensis* (with the prototypical strain Lelystad) and American (subgenus *Ampobartevirus*, species *Betaarterivirus americense* (with the prototypical strain VR-2332). The origin of PRRSV has not yet been determined. European and American PRRSV isolates cause similar clinical signs, but are two different species whose genomes differ by about 40% [204]. Researchers hypothesized that the lactate dehydrogenase-elevating virus (LDV) and PRRSV originated from a common ancestor. It was hypothesized that LDV evolved into PRRSV and that wild boars acted as intermediate hosts [205,206]. LDV was first isolated from laboratory mice with tumors but was later found to be endemic in domestic mouse populations, causing an asymptomatic course of disease. LDV replicates in a subpopulation of macrophages that removes excess lactate dehydrogenases from the blood flow [205,207]. Researchers have been making assumptions about the occurrence of RRRSV since 1912, when 14 wild boars were imported into North Carolina (USA). Some of these animals were infected with a precursor to the current RRRSV [208].

70 years later, the virus evolved independently in the pig population, with a common ancestor [208]. In 1987, a disease with reproductive disorders in sows was identified in Canada and the USA. By 1990 an outbreak of similar clinical signs in pigs was reported in Germany, and by 1991, the virus had spread across Europe [209,210,211]. The etiology of the disease was established in 1993 in the Netherlands when the Lelystad strain of PRRSV was isolated from primary alveolar macrophages (PAM) [209,210,212]. In the USA (1992), a virus with similar clinical manifestations was also isolated. It was named VR-2332. As it later turned out, it is a new species—American PRRSV [213].

American and European PRRSV strains exhibit similar genome organization, with key differences localized primarily in the 5′ and 3′ non-coding regions, as well as in the ORF1a and ORF1b coding regions (proteins Nsp2 and CP4, respectively) [214]. When PRRS virulence was assessed in farrow sows, the European isolate (CBNU0495) demonstrated high replicative activity. Whereas the American isolate (K07-2273) induced more pronounced reproductive disorders in sows, despite reduced replication in fetal cells. Notably, viral load and histopathological alterations showed no correlation [215].

PAMs are considered to be the most reliable biological model for PRRSV replication, as their cell surface expresses the CD163 receptor, essential for mediating virion adsorption. Notably, PAMs remain the only non-genetically modified porcine cells suitable for efficient virus cultivation, despite attempts to utilize alternative cell lines such as blood monocytes or monocyte-derived dendritic cells [216,217,218]. CPE caused by the European PRRSV (Lelystad strain) is typically observed in PAMs 3–4 days post-inoculation, manifesting as cell de-adhesion and cytolysis. The European strain of PRRSV isolated from a domestic pig in China (HLJB1) in 2011, accumulated in PAMs at a titer of 5.0 lg TCID_50_/mL [219]. When assessing the permissivity of macrophages from suckling piglets, young and adult pigs to PRRSV, observations confirmed that viral replication correlated more strongly with the donor’s metabolic state than with age. Although the proportion of macrophages was significantly lower in adult pigs compared to young ones, the percentage of CD163-receptor-positive cells in suckling piglets and young pigs did not differ, consistently reaching 90–100% of the total PAM population [220].

The assessment of the MARC-145 cell line’s permissivity to PRRSV demonstrated its promising potential for primary virus isolation. At an MOI of 0.1 TCID_50_/cell, the titer of the Lelystad strain reached 5.04 lg TCID_50_/mL by 96 h post-inoculation (hpi), indicating high infectious activity. Viral accumulation increased progressively over time, confirming efficient replication in this cell line. Furthermore, the data showed an increasing trend across the first three passages, reaching 5.51 lg TCID_50_/mL [221].

Other cell cultures did not demonstrate high permissivity to PRRSV and were not subsequently used for isolation. In contrast, viral titers declined in both PK-15 and Vero cell cultures with successive passages:

PK-15: Decreased from 5.05 lg TCID_50_/mL (passage 1) to 4.97 lg TCID_50_/mL (passage 3).

Vero: Decreased from 4.61 lg TCID_50_/mL (passage 1) to 4.39 lg TCID_50_/mL (passage 3) [221].

The St. Jude pig lung (SJPL) cell line is a continuous cell line derived from the ciliated epithelium of the porcine respiratory tract, which was used to study PRRSV replication. When infected with the American IAF-Klop strain at an MOI of 1 TCID_50_/cell, SJPL cells exhibited an uncharacteristic CPE 72 h post-inoculation (hpi), differing significantly from values obtained in the MARC-145 culture. Notably, the CPE severity in SJPL cells at 120 hpi was comparable to that in MARC-145 cells at 72 hpi, indicating delayed viral replication kinetics. The PRRSV titer in SJPL cells was significantly different between passages: 3.3 lg TCID_50_/mL at passage 1 and 6.6 lg TCID_50_/mL at passage 5. This increase in titer demonstrates that the SJPL cell line, like MARC-145, is susceptible to PRRSV [222]. However, this method is not yet widely used in the practice of virus isolation due to the difficulty of its implementation in global cell culture collections.

It is also important to note that other cell lines, for example, porcine trophectoderm (PTr2), which express specific markers for porcine trophoblast, also showed moderate accumulation levels when infected with the American PRRSV. Infected PTr2 cells developed CPE by 72 hpi, demonstrating their permissivity to PRRSV. Immunofluorescence assay (IFA) detected viral antigen in infected cells at all examined time points (24, 48, and 72 hpi). Flow cytometry revealed an MOI-dependent infection rate: at an MOI of 1 TCID_50_/cell, the number of infected cells decreased to 4.65% by 72 hpi, while at an MOI of 5 TCID_50_/cell, the number of infected cells was significantly higher (12.80%) [223].

Thus, despite exhibiting strict tropism for specific cells within porcine macrophages, PRRSV can replicate efficiently in several continuous cell lines derived from diverse animal tissues (MARC-145, SJPL, and PTr2) [224]. It is preferable to use PAMs and, less preferably, MARC-145 for PRRSV isolation. It is important to correctly identify PRRSV after each passage. For indirect methods of assessing viral replication, it is better to use quantitative PCR (qPCR) than titration, as with each passage, including final dilution testing, PRRSV tends to reduce viral load and therefore produce unreliable results. A quick guide to PRRSV isolation is shown in Figure 6.

## 9. Porcine Circoviruses

Porcine circovirus infection is a persistent secondary multi-syndromic disease.

Porcine circovirus infections are caused by small, non-enveloped viruses with single-stranded circular DNA that belong to the *Circovirus* genus within the *Circoviridae* family. The first identified species, porcine circovirus 1 (PCV-1), was discovered in 1974 by German researchers as a non-pathogenic contaminant of the PK-15 pig kidney cell line and was not recognized as a pathogen at the time of its discovery [225,226].

In 1982, a group of researchers isolated a circovirus (PCV-2) from infected pigs that was genetically different from the cell culture contaminant [227]. Retrospective analysis of archived sera revealed PCV-2 had circulated long before its identification: in Belgium since 1969, Great Britain since 1970, Northern Ireland since 1973, and Canada and the USA since 1985 [228]. In 1991, Canada reported outbreaks of a severe, previously unknown disease causing high mortality in 5–12 week-old piglets. This condition was subsequently named post-weaning multi-systemic wasting syndrome (PMWS) [229,230]. The etiological agent, porcine circovirus 2 (PCV-2), was isolated in 1997 and confirmed as the cause of PMWS [231].

Porcine circovirus 3 (PCV-3) was first described in 2015 in North Carolina (USA), with increased mortality and decreased fertility reported in sow herds [232].

Porcine circovirus 4 (PCV-4) is a newly emerging virus, with both PCV-4 genomic DNA and specific antibodies detected in swine herds in several provinces in China and South Korea. Although first isolated in 2019 from China’s Hunan Province, retrospective studies identified PCV-4 DNA in serum samples dating back to 2012 [233]. This indicates the long-term persistence of PCV-4 in pigs.

Among porcine circoviruses, PCV-2 remains the most significant veterinary pathogen, associated with a wide spectrum of clinical syndromes. These include PMWS, porcine dermatitis and nephropathy syndrome (PDNS), necrotizing lymphadenitis, granulomatous enteritis, reproductive failure, congenital tremor, pneumonia, and systemic lesions affecting multiple organ systems [29,234].

PCV-2 was first isolated from tissues of PMWS-affected piglets in a PCV-1-free continuous PK-15 cell culture and accumulated at a titer of 4.3–5.5 lg CCID_50_/mL. Since the virus did not exhibit CPE, the detection of PCV-2 replication was registered in IFA [235]. An experiment was also conducted using clones of the PK-15-Cl and PK-15-A2 cell lines to assess PCV-2 viral load via indirect IFA. In the PK-15-Cl clone monolayer, the maximum virus titer reached 8.0 lg CCID_50_/mL after several passages, significantly exceeding the lower titer of 5.0 lg CCID_50_/mL observed in the original PK-15 (ATCC-CCL31) cell line. Moreover, viral accumulation in PK-15-A2 cells never exceeded 2.0 lg CCID_50_/mL [236]. This once again indicates the heterogeneity of PK-15 sublines in terms of virus permissivity, as is the case with CSFV.

PCV-2 isolates are generally considered non-cytopathic. Nevertheless, researchers use diverse cell lines to characterize the replication kinetics of viral isolates. Quantitative assessment is carried out using IFA and qPCR. For example, infection of CPK-NS cells resulted in pronounced cell de-adhesion after inoculation and high viral titers (>4.5 lg TCID_50_/mL). Thus, the suggestion was made that these cells are highly permissive to PCV-2 and would allow the detection of an infectious active virus in biological material even when viral load is very low. PPK-3F (piglet lung fibroblasts) also demonstrated susceptibility to PCV-2 (qPCR at passage 3 revealed higher PCV-2 genome copy numbers in PPK-3F cells (14.8 copies/μL) compared to CPK-NS cells (8.9 copies/μL), indicating greater replication efficiency in PPK-3F cells) [237].

Further analysis showed that PCV2-like CPE could still be induced in cell cultures. It was characterized by increased intercellular spaces and cell elongation in both POMECs (primary porcine oral mucosal epithelial cells) and hTERT-POMECs (human telomerase-expressing POMECs). These changes became apparent 24 h post-inoculation (hpi) and intensified as incubation time increased. Pronounced CPE was reported 72 hpi. PCV-2 titers reached 7.1 lg TCID_50_/mL in POMECs and 6.9 lg TCID_50_/mL in hTERT-POMECs. These values significantly exceed those reported for the PK-15 cell line, which is widely used for circovirus cultivation [238]. The fetal porcine retinal cell line (VR1BL) was assessed for permissivity to porcine circoviruses. Results demonstrated that VR1BL cells are highly susceptible to PCV-2. At MOI of 0.5 TCID_50_/cell, the viral titer in VR1BL culture suspension reached 6.0 lg TCID_50_/mL, which is significantly higher than when cultured in PK-15 (2.0 lg TCID50/mL). VR1BL shows promise as a diagnostic model, especially taking into account replication stability (the “plateau” effect for the virus titer) 24–36 hpi [239].

Despite the widely recognized lack of cytopathogenicity in porcine circoviruses, recent studies demonstrate that PCV-2 can induce characteristic CPE in a number of primary and continuous porcine cell lines. Specified cell lines can serve as effective and valuable models for studying the circovirus replication cycle. Clonal PK-15 sublines are recommended for primary PCV isolation, selected based on their highest permissivity to porcine circoviruses. Viral replication within cell cultures should be identified and quantified using established methods such as IFA or qPCR. A quick guide to PCV isolation is shown in Figure 7.

## 10. Porcine Parvoviruses

Porcine parvovirus infection is a contagious disease that is clinically manifested only in pregnant sows and is characterized by embryonic death.

Porcine parvovirus was first isolated in Germany in 1965 as a contaminant of primary pig cell culture [240]. As early as 1981, frequent resumption of estruation, abortions, a decrease in the number of offspring, and mummified fetuses were noted on pig farms in England. After examining sera, antibodies to porcine parvovirus were detected, leading to the assumption that parvovirus infection had occurred [241]. In the USSR (1984), the disease was discovered by isolating parvovirus from fetuses of aborting sows [242]. In 1985, parvovirus infection was reported in the United States, and a year later in the United Kingdom [243].

Porcine parvovirus infection is caused by viruses from the family *Parvoviridae*, including genera *Protoparvovirus*, *Tetraparvovirus*, *Copiparvovirus,* and *Chaphamaparvovirus* [244]. The classic PPV-1 virus belongs to the *Protoparvovirus* genus. Subsequently, PPV-2 and PPV-3 were assigned to the genus *Tetraparvovirus*. In 2010, signs of PMWS were detected in pigs in the United States, and after sequencing, it was found that the disease was caused by a new type of PPV-4 similar to bovine parvovirus (BPV-2), which belongs to the *Copiparvovirus* genus [245]. In 2013, a new unclassified PPV-5 was identified in the United States. In 2014, DNA samples similar to the parvovirus genome were identified in aborted pig fetuses in China. The virus was named PPV-6 (the *Copiparvovirus* genus) [246,247]. In 2016, PPV-7 belonging to the *Chaphamaparvovirus* genus was detected in the rectal swabs of healthy pigs using metagenomic sequencing [248]. In 2021, a novel parvovirus, designated as PPV-8 (the *Protoparvovirus* genus), was discovered in the respiratory system of infected pigs in China [249]. Thus, the diversity of porcine parvoviruses is great, and there are at least 8 species. This indicates the rapid molecular evolution of this pathogen. At the same time, the current epidemic situation requires permanent laboratory monitoring and improvement of existing inactivated vaccines. This is impossible without isolating relevant strains.

Continuous porcine kidney PK-15 and SK-6 cell lines are most often used to isolate porcine parvovirus (PPV) from biological material. The typical parvovirus-induced cytopathic changes are manifested on permissive cultures no earlier than 72 h post-infection and include cell rounding, increased cytoplasmic granulation, chromatin condensation, and formation of intracellular ‘spider-like’ aggregates [26]. The optimal cultivation period for achieving maximum PPV accumulation in PK-15 cell monolayer is 48–96 h, with a multiplicity of infection of 0.2–1.0 TCID_50_/cell. The virus infectivity, measured by titration in the PK-15 cell line, reaches 5.0 lg TCID_50_/mL, and the hemagglutinating activity, detected in HA, is 7–8 log2 HAU/0.05 mL [244]. It is preferable to use PCR to detect the virus genome in infected cell lines [27]. As a comparative analysis of the virus replication across various cell lines shows, the PK-15 cell line is ranked number one due to its high permissivity. Thus, in PK-15 cells, the titer reached 5.0 lg TCID50/mL, whereas the titer in the continuous porcine embryonic kidney cell line (PPES) was significantly lower, i.e., 3.5–4.0 lg TCID50/mL. At the same time, hemagglutinating activity also varied: in PK-15 cell lines, it was 8–9 log2 HAU/0.05 mL, while in PPES it was 6.0–7.0 log2 HAU/0.05 mL, which demonstrates the high permissivity of PK-15 to PPV and its good prospects for use in the laboratory diagnosis [26].

There is extremely insufficient information about the replication of other porcine parvoviruses (except PPV-1) in cell cultures. Recent data show that primary cell culture can be used to isolate and adapt PPV-5. Thus, the strain Moscow-4060 (2021) was isolated in Russia relatively recently [250].

Thus, porcine cell cultures, primarily the continuous porcine kidney cell line PK-15, are a highly sensitive system for isolating PPV-1 in vitro. Other porcine cell cultures, such as primary and continuous embryonic kidney cells, are also permissive to PPV, but they ensure its accumulation at lower titers compared to PK-15. A quick guide to PPV isolation is shown in Figure 8.

## 11. Porcine Teschoviruses

Teschovirus encephalomyelitis (porcine poliomyelitis, endemic porcine encephalomyelitis) is a contagious viral disease characterized by lesions of the central nervous system. In 1929, the first outbreak of teschovirus encephalomyelitis occurred in Czechoslovakia, characterized by damage to the central nervous system [251,252]. In 1957, a similar disease was observed in Wales, caused by a low-virulence virus and named Talfan disease [253]. In 2009, a high-mortality outbreak of teschovirus encephalomyelitis was recorded in Haiti [254]. Despite the fact that teschovirus encephalomyelitis is a rare disease, it is occasionally found in Japan, Canada, Spain, Brazil, Russia, and China [255,256,257,258,259].

The causative agent is a virus belonging to the genus *Teschovirus* of the *Picornaviridae* family. The main causative agents of the disease are porcine tesoviruses 1 and 11 (PTV-1 and PTV-11). Other teschovirus serotypes, such as PTV2-10 and PTV12-19, cause less pronounced clinical forms of encephalomyelitis [260,261].

Primary cell lines from various porcine tissues, such as kidneys, testicles, and continuous PK-15 and SPEV lines, are used to isolate PTV in the laboratory [260].

Strain KNM, isolated in western Slovakia, is a good example of successful PTV isolation. After nine passages in naturally susceptible animals with intracerebral infection, this strain demonstrated high infectivity, reaching a titer of 4 lg LD_50_/g, which indicates its ability to actively replicate in the nervous tissue of pigs. After four passages in PK, the KNM strain reached a titer of 5–6 lg TCID_50_/mL, demonstrating its high adaptability to various culture systems [262].

During PTV cultivation in PK-15 cell culture, CPE appeared after 48 h, and the cell monolayer destruction occurred within the next 48–72 h. A subsequent three to five viral passages in cell cultures further accelerate CPE manifestation, thereby increasing the viral load. At the same time, virus strains adapted to replication in cell culture, as a rule, reach titers of 6–7 lg TCID_50_/mL, which confirms their high infectious activity and significant potential for further use for diagnosis and research purposes [260].

The successful isolation of PTV field isolates was observed in a continuous ST cell line; CPE was observed 96–120 h post-infection, and virus accumulation accounted for 3.0 to 3.75 lg TCID_50_/mL. Adaptation of the isolates was studied in PSGK-30 to obtain virus-containing culture material with high infectious activity. The accumulated viral titer reached 6.0–7.5 lg TCID_50_/mL, and CPE appeared 24–36 h later [263].

Summarizing the data given, various primary and continuous porcine cell cultures, including kidney and testicle cells, as well as embryo kidneys, can be recommended to isolate PTV from pathological material. CPE is characterized by small foci of rounded, shrinking cells. It should be borne in mind that PTV-induced CPE in sensitive cultures is manifested by monolayer destruction, which intensifies with longer incubation times and a higher number of serial virus passages. A quick guide to PPV isolation is shown in Figure 9.

## 12. Porcine Coronaviruses

Coronaviruses represent a diverse group of porcine pathogens belonging to the family *Coronaviridae*. These are enveloped, positive-sense, single-stranded RNA viruses that cause a number of economically significant diseases, including transmissible gastroenteritis, respiratory coronavirus infection, epidemic diarrhea, porcine deltacoronavirus, acute diarrhea syndrome, porcine hemagglutinating encephalomyelitis, and others. These viruses differ in their genetic and antigenic properties, tissue tropism, and clinical manifestations of the diseases they induce [264,265].

***Transmissible gastroenteritis virus*** (TGEV) is a highly contagious disease caused by a virus belonging to the genus *Alphacoronavirus,* causing ulcerative enteritis, dehydration, and a rise in mortality rate in newborn piglets up to 100% [266]. TGEV was first discovered in the USA in 1946. After that, reports began to appear of its occurrence in Europe, Asia, and Africa. This causes serious economic losses to pig farming [267,268,269,270].

The process of virus isolation under laboratory conditions is possible using a monolayer culture of porcine kidney cells (PK) [271]. In this case, the cytopathic effect resulting from the virus cultivation appears 3–7 days afterwards. Virus replication is accompanied by cell rounding and enlargement, syncytium formation, and subsequent cell detachment into the growth medium. However, field isolates demonstrate reduced cytopathogenicity, which requires a series of “blind” passages before typical morphological changes in cells become apparent [272].

The highest concentration of virus particles can be achieved by roller cultivation of the virus in the SPEV line (up to the titer of 7.0–8.0 lg TCID_50_/mL). Even higher titers of TGEV are observed in suspended porcine kidney cell line (PPK-66b)—8.0–9.0 lg TCID_50_/mL, which confirms the good potential of this cell line for diagnosis purposes [273]. In the SPEV and ST monolayer, TGEV titer achieves 7.75 lg TCID_50_/mL. Signs of cell degeneration included cell shrinkage, disappearance of the nuclear contour, and cell death occurring within 24–36 h after the virus inoculation [274].

The ability of TGEV to replicate has also been proven in primary PAM cell culture. The virus titers were 5.6–6.7 lg TCID_50_/mL. In vitro experiments have shown that pig blood monocytes are not sensitive to the studied pathogen [275]. The possible interaction of TGEV with intestinal macrophages at the initial stages of infection requires further in vitro tests [276].

***Porcine respiratory coronavirus*** (PRCV) has evolved as a result of TGEV’s natural mutation, i.e., a large deletion at positions 621–681 nt in the S gene and minor deletions in the 3/3a and 3-1/3b genes, first discovered in Belgium in 1984, and in the USA in 1990 [277]. These mutations alter the viral tissue tropism: unlike TGEV, which primarily replicates in the epithelial cells of the small intestine and causes severe diarrhea in piglets, PRCV exhibits selective tropism for the respiratory epithelium and induces mild respiratory infection clinical signs without diarrhea. The emergence and spread of PRCV in swine populations has gradually reduced the incidence of transmissible gastroenteritis observed in 1984 [278,279].

Isolates exhibiting CPE were sequentially isolated in primary ST over four passages. The infectious activity was tested in PCR with an amplification threshold cycle (Ct) ranging from 18.7 to 23.9 [280].

In vitro replication of the virus exhibiting a cytopathic effect was also observed in porcine kidney cell culture (LLC-PK1); the infectious activity observed after 24 h-incubation following culture inoculation was 6.0 lg TCID_50_/mL [281].

***Porcine epidemic diarrhea virus*** (PEDV), belonging to the genus *Alphacoronavirus*, was first discovered in the UK in 1971 [282]. In 1983, PEDV spread to Japan [283]. In 2006, six strains were isolated from fecal samples in China [284]. Subsequently, despite the initial low mortality rate, a PED outbreak occurred in China again in 2010 with high virulence, leading to the death of pigs in 100% of cases. The CH/S strain was used as a virulent strain to evaluate vaccine effectiveness in China [285]. In the United States, Canada, and Mexico in 2013, for the first time, the disease caused the death of 8 million pigs [286,287,288].

The cytopathic virus showed permissiveness in Vero cell culture (African green monkey kidney). CPE, characterized by vacuolization, syncytia formation with 5–10 nuclei, and cell fusion, was observed in Vero cells as early as the first virus passage. Starting from passage 5, PEDV induced the complete fusion of Vero cells’ monolayer, resulting in total destruction within 24 h post-infection. During the first 10 serial passages, virus titers ranged from 2.7 to 5.3 lg TCID_50_/mL [289,290]. No PEDV replication was detected in the other cell lines tested [291].

***Porcine deltacoronaviruses*** (PDCoV). The disease is characterized by diarrhea, vomiting, dehydration, and death of newborn piglets. After its discovery in Hong Kong in 2012, the International Committee on Taxonomy of Viruses (ICTV) established the new genus, *Deltacoronaviruses*, based on a whole-genome analysis of isolates from pigs δ-CoV (CoV HKU15), with the interesting finding that a partial sequence of porcine deltacoronavirus genes had been identified in Chinese ferrets as early as 2006 [292]. Since 2014, PDCoV has been reported in the United States [293]. In 2014, the KNU14-04 strain was identified in pigs in South Korea [294].

LLC-PK and ST cell lines are used for PDCoV isolation and replication [293,295]. Cell culture-adapted strains of PDCoV achieved the following titers: USA/IL/2014 (Passage 11), 4.7 lg TCID_50_/mL; Michigan/8977/2014 (Passage 15), 6 lg PFU/mL [293]. For efficient replication of PDCoV in LLC-PK and ST cells, the culture medium should be supplemented with 10 μg/mL trypsin and 1% pancreatin.

***Swine* *acute diarrhea syndrome* virus** (SADS-CoV) was first reported in China in 2017 and classified as a representative of the genus *Alphacoronavirus* [280]. Infection with the new coronavirus is characterized by acute diarrhea and vomiting, with mortality among piglets under 5 days old being 90% [296]. Researchers suggest that the virus originates from *Rhinolophus* spp., bats living near local outbreak sites, and has a similar affinity for HKU2, a bat coronavirus [297].

The production of synthetic recombinant SADS-CoV virus made it possible to determine its cell tropism range in vitro. It was established that SADS-CoV is able to effectively replicate in a wide range of animal cell lines, including human liver, lung, and intestinal cells. The most optimal models were Vero (CCL-81) and LLC-PK1. Virus titers range from 4.0 to 7.5 lg TCID_50_/mL. The infection of sensitive cells with SADS-CoV is followed by CPE, which includes syncytium formation and cytoplasm vacuolization and manifests itself 24 h post-infection [298].

***Swine enteric coronavirus*** (SeCoV). The origin of SeCoV is still unclear. In 2020, researchers conducted a retrospective assessment of positive samples for PEDV in pigs, and the analysis showed that outbreaks of diarrhea caused by a recombinant virus periodically occurred in Spain in the 1990s (SeCoV-Italy/213306/2009-KR061459) [299]. The origin of SeCoV is most often associated with coronaviruses from bats and birds [265].

Mixed infections caused by several SeCoV representatives are widespread on pig farms all over the world [300]. SeCoV leads to the apoptosis of infected cells, while the apoptosis regulation mechanisms vary depending on the type of cells and the infectious process stage [279]. Strain SeACoV/CH/GD-01/2017, related to human coronavirus HKU2, was isolated in Vero cells, inducing syncytium formation 48 h post-infection (hpi), starting from passage 2. SeACoV RNA was detected in the supernatants between passages 2 and 8, and the virus titer in passage 8 reached 6 lg PFU/mL [301].

***Porcine hemagglutinating encephalomyelitis virus*** (PHEV) belongs to the genus *Betacoronavirus* and infects piglets under 3 weeks of age [302,303,304]. The first clinical outbreak of the disease was reported in Ontario, Canada, in 1957 [305]. The disease was systematically recorded until 1961 and was characterized by high mortality, vomiting, anorexia, dehydration, and severe progressive exhaustion in weaned piglets [305,306]. Despite the long absence of reports of the disease, the virus was detected in adult pigs with signs of respiratory disease at a fair in the United States in 2015 [307].

In order to cultivate and titrate the PHEV strain VW 572, primary cell cultures and continuous porcine thyroid cell lines, as well as PK-15, SK-6, and ST, are used. As demonstrated by the cytopathic effect, immunofluorescence, and hemagglutination, porcine thyroid cells and primary porcine kidney cells are the most susceptible to PHEV [308].

Strain 67N, adapted to the nervous tissue of newborn mice, effectively replicates in porcine kidney cells of the SK-K line, resulting in syncytia formation and cell detachment. During serial passages of the virus in SK-K cells, CPE developed 48 hpi, and the PHEV titer reached more than 7 lg PFU/mL. PHEV antigen was detected in the cytoplasm of infected SK-K cells using IFA [309]. However, the permissiveness of this cell culture for field PHEV isolates has not been considered, which requires further research.

There are few studies on the permissiveness of cell cultures to PHEV; however, it has been established that a differentiated culture of epithelial cells from the respiratory tract, tonsils, nasal mucosa, and lungs of SPF pigs is susceptible to PHEV replication, developing marked cytopathic effects within 36–48 h post-infection [310].

Summarizing the given data, it can be concluded that porcine coronaviruses are characterized by significant genetic and antigenic diversity. To isolate and study the biological properties of porcine coronaviruses, including their replication characteristics and cytopathic effects, primary porcine kidney cell culture (for TGEV and PHEV), and the continuous porcine kidney cell line LLC-PK1 (for PRCV, PEDV, PDCoV, SADS-CoV), as well as a continuous kidney cell line from the green monkey Vero (for PEDV, SADS-CoV, SeCoV), are widely used. The virus accumulation in permissive cell cultures allows for obtaining satisfactory virus titers (between 4.0 and 8.0 lg TCID_50_/mL). For the efficient replication of certain swine coronaviruses in cell culture, it is required to add exogenous proteases (such as trypsin or pancreatin) to the culture medium. It has been proven that these enzymes cleave S1 and S2 subunits of the S-protein (Spike), enhancing the coronavirus’s endocytosis into cells, and can also lead to increased permissivity and expansion of specific cellular hosts [311].

Cell cultures serve as a valuable tool for studying the pathogenesis of swine coronavirus infections, developing effective laboratory diagnostic methods, and assessing genetic interactions between different isolates. A quick guide to porcine coronavirus isolation is shown in Figure 10.

## 13. Swine Vesicular Diseases

Vesicular syndrome is characterized by the formation of vesicles (blisters) of various localizations on the skin and mucous membranes. The most significant diseases in this group include foot-and-mouth disease, swine vesicular disease, vesicular exanthema of swine, vesicular stomatitis, and senecavirus infection in pigs. Despite certain differences in etiology, these diseases have similar clinical signs and require differential diagnosis.

The most dangerous for pigs is ***foot-and-mouth disease virus (FMDV)***. It is a serious epidemic disease that threatens livestock production. In 1546, Hieronymus Francastorio described an unusual disease in cattle near Verona, Italy [312]. F. Loeffler and P. Frosch, in 1897 [313], presented a brief report on research on salivary and maxillary diseases. They described the causative agent of FMD as a filtering agent [313]. In 1922, several serotypes of FMDV were identified. In France, H. Valley and H. Carre described serotypes O and A, while in 1926, O. Waldmann and K. Trautwein described serotype C [314,315]. Three new serotypes were also identified in South Africa in 1948 (SAT 1, 2, and 3) [316]. In the early 1950s, the seventh serotype, Asia 1, was isolated in India and Pakistan [317]. Currently, 6 of 7 serotypes of FMDV are circulating. Serotype C has been considered eradicated.

For pig farms around the world, the disease causes great losses. In 1997, serotype O originated in Taiwan and caused the death of more than 3.5 million pigs [318]. In the UK, 2030 cases were registered in 2001 [319]. In 2015, 180 pig farms in South Korea suffered a total financial loss of US $25.2 million [320].

The FMD clinical manifestation in pigs involves fever, depression, anorexia, and the formation of vesicles in the mouth, on the tongue, snout, and extremities, especially on the coronary band and interdigital cleft [321]. Comparative studies of the domestic and wild pig susceptibility to FMDV A-24 Cruzeiro demonstrated that pigs were highly susceptible both to intradermal infection and contact infection due to the virus transmission from infected animals. In the domestic pigs, the FMD clinical signs developed in a shorter timeframe after contact with the diseased animals (24 h) compared with the wild pigs (48 h). Nevertheless, the dynamics of the virus accumulation and excretion were similar in both groups of animals, with maximum virus titer in oral swabs collected on days 2–4 post-infection [322].

The most sensitive system for FMDV isolation is primary bovine thyroid cells (BTY), but they are seldom used due to the difficulties in sourcing the tissues and their relatively short lifespan [323].

To isolate and quantify the swine FMDV, porcine cell cultures are generally used: PK and IB-RS-2. During replication in these cell cultures, the FMDV viral load reaches 8.5 lg TCID_50_/mL [324]. To detect the FMDV reproduction in certain porcine cells, samples of the tissues were collected where visible lesions were most often detected (i.e., tissue samples from the dorsal part of the soft palate) in order to derive and culture the cells. The virus titer on the dorsal soft palate (DSP) cell culture ranged from 5.2 to 6.7 lg TCID_50_/mL [325].

Cell cultures of non-porcine origin can also be permissive to FMDV. The BHK-21 cell culture-adapted FMDV strain, O UKG 34/2001 BTY1/F1, derived from clinical samples of infected pigs, can achieve a titer of up to 7.6 lg TCID_50_/mL [326]. The fetal goat tongue cell line (ZZ-R 127) demonstrated good permissivity to replication of various FMDV strains, with accumulation titers at hour 48 ranging from 5.9 to 8.6 lg TCID_50_/mL. The virus CPE was visible as early as 24 h after the virus inoculation. In comparison, the titer of the virus accumulation on IB-RS-2 at 48 h after infection ranged from 4.8 to 6.7 lg TCID_50_/mL [327].

***Swine vesicular disease virus* (SVDV)** belongs to the family *Picornaviridae*, genus *Enterovirus*.

Outbreaks caused by the virus were reported in Italy in 1966, in Hong Kong in 1971, and in Great Britain in 1972 [328,329,330].

The causative agent of SVD is a cytopathogenic virus that is prone to a rapid replication cycle. When observing the changes after the PK cell infection, vacuolar dystrophy of cytoplasm and intense nucleus basophilia were reported as early as the fourth hour. After eight hours, karyopycnosis followed by karyorexis was observed in the cytoplasm. The complete degeneration of the cellular monolayer was reported after 18–24 h. Therefore, replication of the SVDV in cell cultures can be analyzed on a real-time basis by detecting the CPE and recording changes in the cell proliferation index [331]. However, despite the presence of SVDV-replication in primary porcine kidney cell culture, it was not possible to adapt SVDV isolates to serial replication during passage dynamics [332,333].

Nevertheless, SVDV is able to adapt to other cultures. Thus, after SVDV infection with PK-15 culture, CPE was detected in the cell culture after 24–48 h. Light-reflecting rounded cells typically appeared throughout the monolayer with the subsequent formation of grape-like clusters, which then detach from the glass surface [333]. The virus also reaches high titers in the continuous IB-RS-2 and PSGK-30 cell lines. During the first passages in cell culture, 100% CPE most often appears 48 h after infection, and by the third passage, complete monolayer destruction is reduced to 18 h, and the virus is accumulated in the material over the next five passages in titer ranging from 6.98 to 7.8 lg TCID_50_/0.05 mL [334].

The virus demonstrated high replicative capacities in porcine cell cultures derived from NSK and NPTr; the virus titer amounted to 7.50 and 7.24 lg TCID_50_/mL, respectively [149].

Primary cell cultures of calf kidney, chicken embryonated eggs, continuous sheep adrenal cell culture, and BHK-21 were found to be non-permissive to the SVDV [335].

***Vesicular exanthema of swine virus*** (VESV) is an RNA virus of the genus *Vesivirus*, family *Caliciviridae*. The disease was first discovered in California in 1932 and spread throughout the United States until 1952, before a ban on feeding pigs with untreated meat waste was introduced [336]. American researcher Traum demonstrated the disease’s distinct etiology and established its viral nature in 1936 [337,338]. The VESV was first discovered and described by Madin and Traum in 1953. Currently, 13 serotypes of VESV are identified, among which types B and D are pathogenic exclusively for pigs, while the remaining serotypes are capable of causing infection in horses and sea lions [339].

Effective systems for vesicular exanthema virus isolation from the pathological material include primary PK cell culture and continuous IB-RS-2 cell line [339]. The dynamics of propagation and the nature of CPE in VESV are the same as in SVDV. Therefore, virus typing should be carried out after each passage in PCR.

The group of diseases with porcine vesicular syndrome also includes ***vesicular stomatitis*** (VSV). It was first identified after an outbreak in the USA in 1916 [340]. The disease is endemic from South America to the south of Mexico, and it is represented by two serotypes, named after the American states where they were identified: New Jersey (VSNJV) and Indiana (VSIV). These serotypes mainly cause the disease in horses and cattle, but pigs can also exhibit clinical signs [341,342]. The virus has zoonotic potential and can be transmitted to humans. Between 2004 and 2006, there were 751 outbreaks of vesicular stomatitis reported in the US, and in 2019, there were 1144 reported outbreaks in eight US states [343,344].

Vesicular stomatitis is caused by an RNA virus of the genus *Vesiculovirus*, family *Rhabdoviridae*. In general, VSV has a wide range of specific cellular hosts and can be replicated in a variety of cell cultures. This virus usually does not cause difficulties in selecting permissive cell cultures and cultivation conditions. This is due to the fact that low-density lipoproteins (LDL), which are present in most cells of the animal and human organism, act as a receptor for the attachment of VSVG-protein on the surface of susceptible cells [345].

Continuous African green monkey kidney cell line (Vero), baby hamster kidney cell line (BHK-21), bovine kidney cell line (MDBK), and primary PK are used for VSV isolation and cultivation. The highest viral load was demonstrated at 9.8 lg TCID_50_/mL in BHK-21 [346].

***Senecavirus A*** (SVA), an RNA virus of the genus *Senecavirus*, *Picornaviridae* family, has been relatively recently included in the group of porcine diseases with vesicular syndrome. In 2002, the SVV-001 strain was discovered by chance in cell culture. Then, it was found again in infected pigs in 2008 and 2012 with idiopathic vesicular disease [347]. SVA has zoonotic potential like VSV and can be transmitted to humans. Since 2015, there has been a significant increase in the number of senecavirus outbreaks in pigs in various countries of the world, including the USA, Canada, Thailand and Brazil [348,349,350,351,352].

Senecavirus causes a representative cytopathic effect in PK-15, manifested by the cell rounding, reduction in size, and detachment from the substrate. The senecavirus field isolates obtained in China (SVA-GD5-2018 and SVA-GDSZ-2018) demonstrated a high replication rate in PK-15 cells, and they were capable of reaching as high titer of 10.5 lg TCID_50_/mL within 24 h after the monolayer infection [353]. Thus, the universal PK-15 cell line is the preferred model for virus isolation.

To summarize, we can conclude that for laboratory diagnostics and study of the biological properties of the viruses causing porcine diseases with vesicular syndrome, the continuous cell lines derived from kidney tissues of naturally susceptible and laboratory animals, including pigs (IB-RS-2, PK-15), rodents (BHK-21), and monkeys (Vero) are used. These viruses are characterized by a wide range of specific cellular hosts, rapid accumulation dynamics, and the achievement of high titers (up to 10.30 lg TCD_50_/mL) within 1–2 days after infection, and they do not present significant difficulties to isolation in cell cultures. However, due to similar CPEs and lack of selective culture, viruses that cause porcine vesicular disease should always be tested for PCR using biological and cultural materials for differentiation, including for cross-contamination. A quick guide to isolating porcine viruses causing vesicular syndrome is provided in Figure 11.

## 14. Other Viruses

***Swine pox virus (SPV)*** is the only member of the genus *Suipoxvirus*, which belongs to the subfamily *Chordopoxvirinae*, family *Poxviridae*. This virus causes a disease of pigs characterized by the formation of a papular–pustular rash on the skin and mucous membranes [25,354].

Primary and continuous pig kidney and testicular cell cultures are commonly used for SPV isolation and cultivation. PK-15 cell line (ATCC line) is widely used for SPV replication and exhibits infectious activity of 6.5 lg TCID_50_/mL [355]. Primary STs were also found to support successful SPV isolation and accumulation: the titer was 6.0 lg TCID_50_/mL at the fifth passage [356].

It should be noted that the SPV replicability in cell cultures derived from tissues of other animal species is significantly limited as compared to cells of porcine origin. Thus, attempts to adapt SPV to MDBK cells, continuous bovine kidney cell line, were unsuccessful [357].

***Porcine adenoviruses*** (PAdVs) belong to the genus *Mastadenovirus*, family *Adenoviridae*, and include three species: PAdV-A, PAdV-B, and PAdV-C [358,359,360,361,362]. Adenoviruses are non-enveloped viruses with double-stranded DNA that are able to infect both dividing and non-dividing cells, which makes them promising vectors for delivering genetic material to target cells [358]. The first PAdV isolates were obtained from a rectal swab collected from the piglet with diarrhea by its inoculation on primary PK in 1964 [363]. Transformed cell lines of various tissue origins, including VIDO R1 (porcine retina cells transfected with human adenovirus E1 genes), PK-15, ST and VR1BL (porcine retina), are used for PAdV cultivation in the laboratory. PAdV accumulates to the highest virus titer (up to 8.5 lg TCID_50_/mL) when it is reproduced in VR1BL cells [364,365].

***Torque teno sus viruses* (TTSuV)**, another group of viruses belonging to the fmily *Anelloviridae*, are widespread among domestic pigs and wild boars and cause Torque-teno syndrome. TTSuV are small viruses (up to 50 nm) with single-stranded circular DNA and are divided into two genera: *Iotatorquevirus* (including the species TTSuV1a and TTSuV1b) and *Kappatorquevirus* (species TTSuVk2a and TTSuVk2b) [366,367]. TTSuV prevalence in pig populations is determined using molecular methods, in particular with PCR [368,369]. According to the literature data, there are no virological methods for virus isolation and cultivation.

Inclusion body rhinitis is caused by porcine cytomegalovirus (***porcine herpesvirus type 2***, SuHV-2), which belongs to the subfamily *Betaherpesvirinae*, family *Orthoherpesviridae*, and causes latent infection in pigs, sometimes manifested by rhinitis with inclusion bodies in piglets and reproductive disorders in sows [370,371]. SuHV-2 is detected in porcine ovarian tissues, follicular fluid, and oocytes using real-time PCR [372]. According to the literature, there are no virological methods for isolating and cultivating this virus.

***Porcine gamma herpesviruses*** (PLHV-1, PLHV-2, and PLHV-3), belonging to the genus *Macavirus* of the subfamily *Gammaherpesvirinae*, family *Orthoherpesviridae*, are highly prevalent in domestic and wild pig populations, but their significance for porcine pathology remains poorly understood [373,374,375]. No virological methods are described; however, it is recommended to use virus isolation methods similar to those used for PRV isolation.

***Menangle virus***, a member of the *Pararubulavirus* genus within the Paramyxoviridae family, is an important RNA virus. This virus, identified in 1997 in Australia, can cause significant reproductive issues in sows. It primarily impacts pregnant sows, causing embryonic and fetal death at different stages of gestation. In immature pigs and non-pregnant sows, the infection is typically asymptomatic [376]. The description of Menangle virus isolation methods using cell cultures has not been publicly released in readily available sources up to this moment.

***Porcine encephalomyocarditis*** (EMCV), caused by a *Cardiovirus*, and ***porcine kobuvirus*** (PKoV) are members of the *Picornaviridae* family and are associated with infectious diseases in pigs. EMCV is a virus that can cause severe health issues in pigs, particularly myocarditis and encephalitis in piglets, leading to sudden death due to cardiac failure, and reproductive problems in sows. To isolate EMCV, the continuous cell lines BHK-21, HeLa (human cervical carcinoma), and Vero are used, as the virus can accumulate in these cell cultures to titers of up to 8.5 lg TCID_50_/mL [377,378]. When PKoV was cultured on RD (human rhabdomyosarcoma) cells, no significant CPE was observed, and viral RNA levels in the medium declined over serial passages. The potential for PKoV replication in Vero and HeLa cell lines requires further investigation [379,380].

***Porcine sapovirus*** (PSaV) is a non-enveloped virus with a single-stranded, positive-sense RNA genome, belonging to the genus *Sapovirus* within the *Caliciviridae* family [381]. PSaV replication in LLC-PK pig kidney cells is dependent on the presence of bile acids. Sapovirus infection of LLC-PK cells leads to a decrease in transepithelial electrical resistance and an increase in paracellular permeability in the early stages of the infection process [382,383]. Transient expression of occludin in nonpermissive CHO cells conferred susceptibility to PSaV, but only for a limited time [384,385].

***Hepatitis E virus* (HEV)** is a single-stranded, positive-sense RNA virus, classified within the genus *Hepevirus* of the family *Hepeviridae* [386]. The difficulty of cultivating hepatitis E virus (HEV) in cell culture limits research into its replication mechanisms in vitro, necessitating reliance on animal models and molecular biology approaches.

***Influenza C*** virus, isolated from pigs, is classified within the *Orthomyxoviridae* family and specifically belongs to the genus *Gammainfluenzavirus* [387,388]. The virus is isolated in ST cells, where a CPE develops by 3 days post-infection, reaching a peak titer of up to 6.9 lg TCID_50_/mL. Swine influenza C virus (strain C/OK) can replicate in MDCK, MARC-145, HRT-18G (human rectal tumor), and A549 (human lung carcinoma) cell lines [389,390].

***Atypical porcine pestivirus virus*** (APPV), belonging to the *Pestivirus K* species, is not a cytopathic virus and can be detected by IFA or PCR [391]. The ability of this virus to replicate in continuous porcine kidney cell lines was noted. M. Beer et al., for example, observed the APPV replication in porcine embryonic kidney cell culture (SPEV) [392]. Virus replication with qPCR detection was also observed after three “blind” passages in PK-15 [393]. The virus RNA was also found in the supernatant after APPV cultivation in SK-L cell lines [394].

***Porcine orthorubulavirus*** causes blue eye disease in pigs and belongs to the genus *Orthorubulavirus* and the family *Paramyxoviridae*. The disease is mainly found in central Mexico [395]. The virus replication has been noted in sheep choroid plexus cultures [396]. A study of the permissivity of porcine (nasal concha, PK, IB-RS-2, PK-15, choroid plexus), bovine (nasal concha, kidney, testicles) and monkey (Vero, green monkey kidney) cell cultures showed that high titers of porcine orthorubulavirus were recorded in porcine choroid plexus—9.0 lg TCID_50_/mL; PK-15 and IB-RS-2—7.0 lg TCID_50_/mL. In Vero, the titer was 5.0 lg TCID_50_/mL [397].

***Nipah virus*** (NiV) belongs to the genus *Henipavirus* of the family *Paramyxoviridae*, and this is a zoonotic virus with a high mortality rate in humans [398]. In pigs, this disease manifests as a respiratory syndrome and clinical signs of damage to the central nervous system [399]. It has been established that the first cases of human Nipah virus infection were associated with pig contact. It has also been found that NiV can be isolated from primary porcine concha and oropharynges [400]. The brain microvascular endothelial cells are the main target cells for NiV [401,402]. The NiV replication has been noted in Vero [403,404]. Within 120 h, an increase in the number of NiV RNA in the supernatant of porcine stable kidney cells (PS) was observed from 3.6 to 8.3 lg PFU/mL [405]. NiV CPE was observed 48 h after inoculation in primary porcine bronchial epithelial cells [406].

In summary, viruses from multiple taxonomic groups—including DNA viruses (*Adenoviridae*, *Anelloviridae*, *Orthoherpesviridae*) and RNA viruses (*Picornaviridae*, *Paramyxoviridae*, *Caliciviridae*, *Hepeviridae*, *Orthomyxoviridae*, *Flaviviridae*)—serve as etiological agents for diverse porcine infectious diseases. Primary and continuous cell cultures derived from porcine tissues (kidney, testes, retina) and other species (hamsters, monkeys, dogs) are utilized for virus isolation and the characterization of biological properties. Many of the studied viruses exhibit a distinct CPE and efficient replication in permissive cell lines, achieving titers of up to 8.5 lg TCID_50_/mL. Currently, optimal in vitro culture systems remain unavailable for several viruses, including TTSuV, SuHV-2, PLHVs, PKoV, PSaV, and HEV. This limitation significantly impedes the investigation of their replication cycles and pathogenesis mechanisms.

## 15. Conclusions

Thus, the pathogens of the porcine viral diseases can be isolated both from homologous (porcine cell cultures) and heterologous (embryonated chicken eggs, continuous green monkey kidney cell lines (Vero, MARC-145, MA-104), baby hamster kidney cell line BHK-21, etc.) biological models.

Cell cultures serve as a valuable tool for the laboratory diagnosis of swine viral infections; however, result interpretation requires consideration of species- and tissue-specific origins of cell lines as well as unique replication characteristics of individual viruses.

The development of NGS (next-generation sequencing) and metagenomics analysis technologies may contribute to the discovery of new porcine viral pathogens. This requires systematic studies of nasopharyngeal swabs, rectal smears, blood, and nervous tissue from pigs, both healthy and those with clinical signs. Understanding the viral nature of swine pathology and subsequent biotechnological research (development of diagnostics and vaccines) will contribute to intensifying pig farming and enhancing food security in countries producing pork.

The authors’ findings, combined with synthesized data from other research groups, can serve as a foundational framework for laboratory practice, facilitating the optimization of culture conditions, obtaining field isolates, and strain adaptation.

## Figures and Tables

**Figure 1 microorganisms-13-02658-f001:**
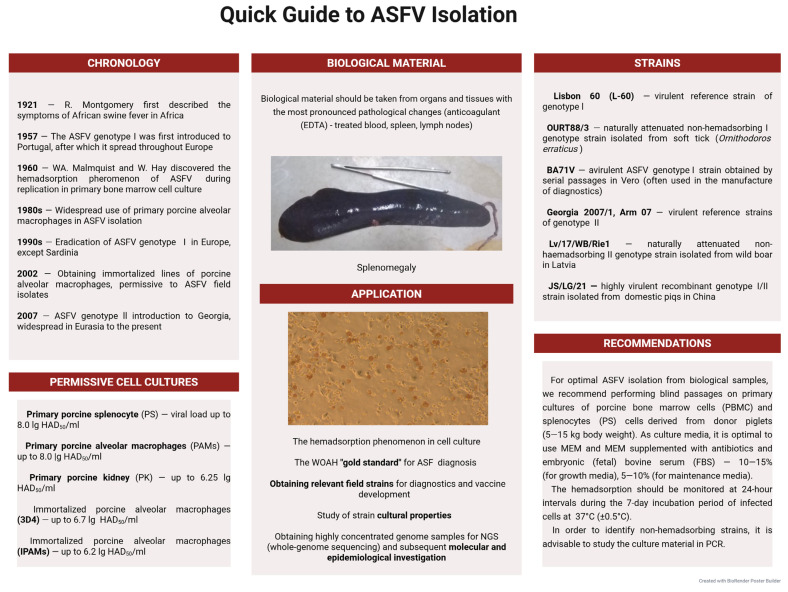
A brief history of the development of virological methods for ASFV isolation and practical recommendations for laboratory personnel. The guide was designed using BioRender Poster Builder.

**Figure 2 microorganisms-13-02658-f002:**
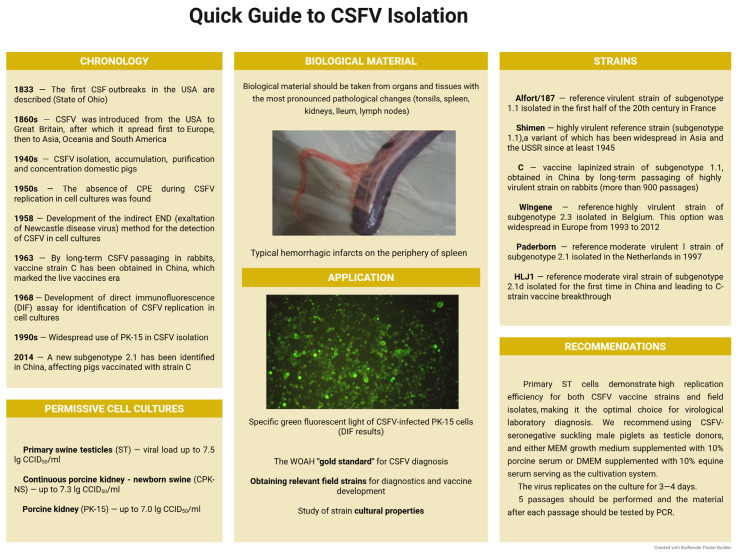
A brief history of the development of virological methods for CSFV isolation and practical recommendations for laboratory personnel. The guide was designed using BioRender Poster Builder.

**Figure 3 microorganisms-13-02658-f003:**
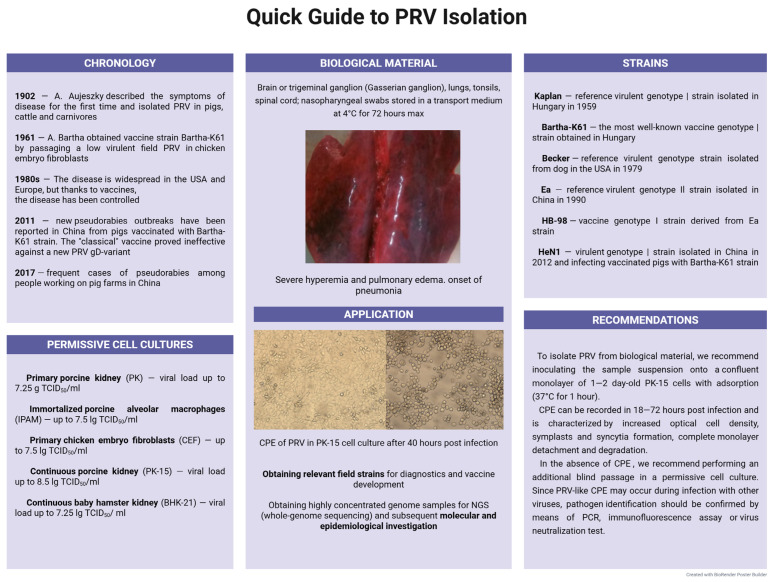
A brief history of the development of virological methods for PRV isolation and practical recommendations for laboratory personnel. The guide was designed using BioRender Poster Builder.

**Figure 4 microorganisms-13-02658-f004:**
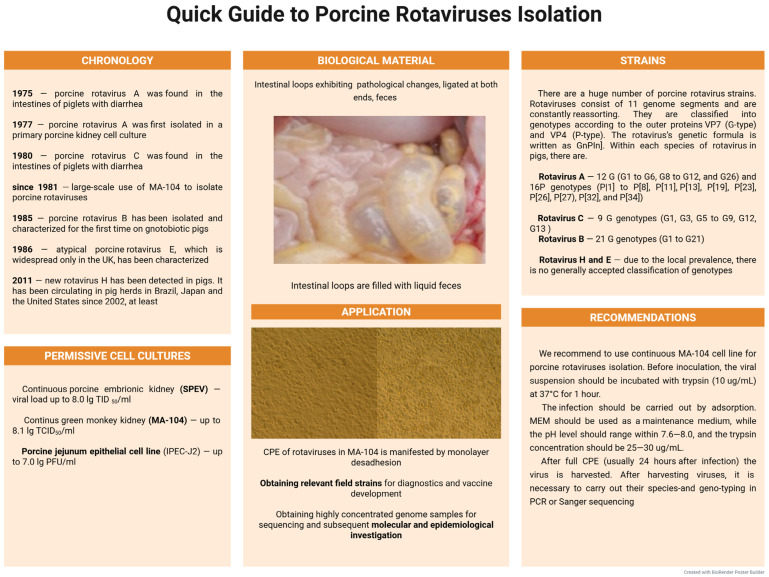
A brief history of the development of virological methods for porcine rotavirus isolation and practical recommendations for laboratory personnel. The guide was designed using BioRender Poster Builder.

**Figure 5 microorganisms-13-02658-f005:**
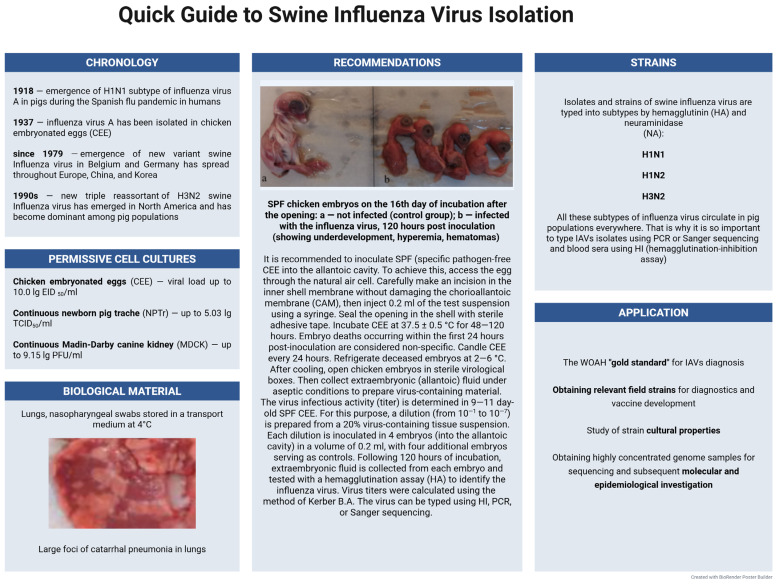
A brief history of the development of virological methods for swine influenza virus isolation and practical recommendations for laboratory personnel. The guide was designed using BioRender Poster Builder.

**Figure 6 microorganisms-13-02658-f006:**
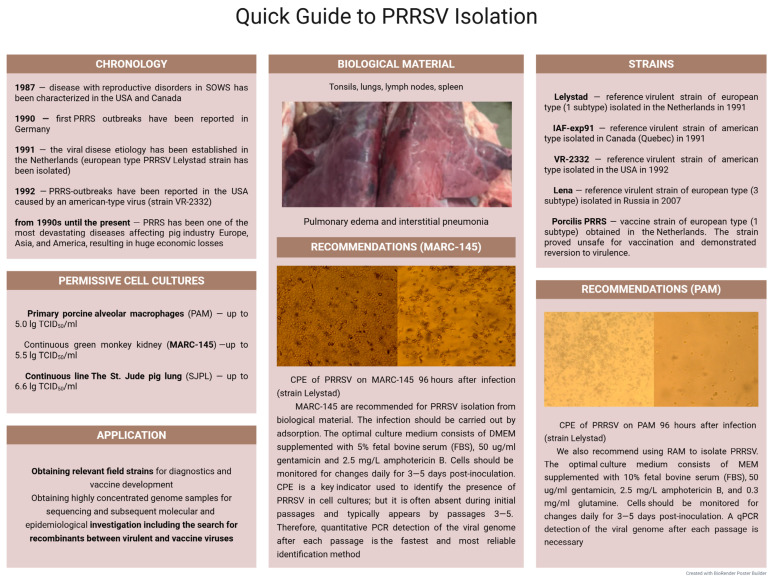
A brief history of the development of virological methods for porcine reproductive and respiratory syndrome virus isolation and practical recommendations for laboratory personnel. The guide was designed using BioRender Poster Builder.

**Figure 7 microorganisms-13-02658-f007:**
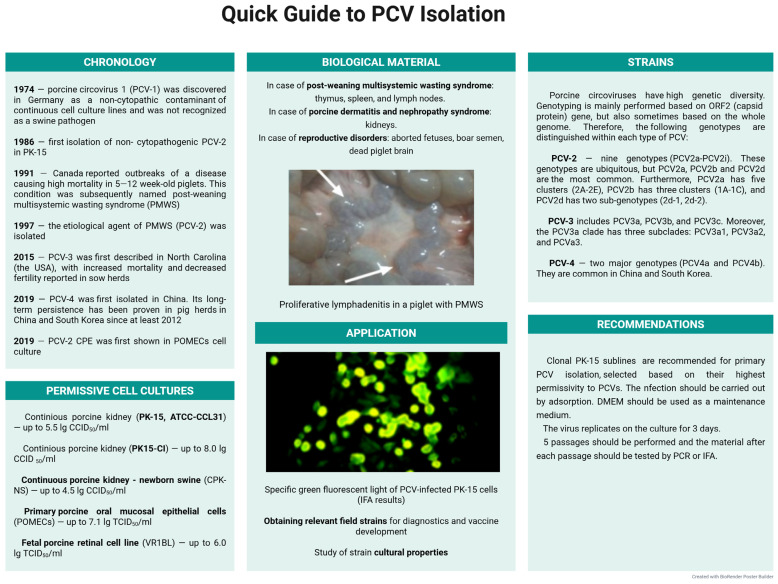
A brief history of the development of virological methods for porcine circoviruses isolation and practical recommendations for laboratory personnel. The guide was designed using BioRender Poster Builder.

**Figure 8 microorganisms-13-02658-f008:**
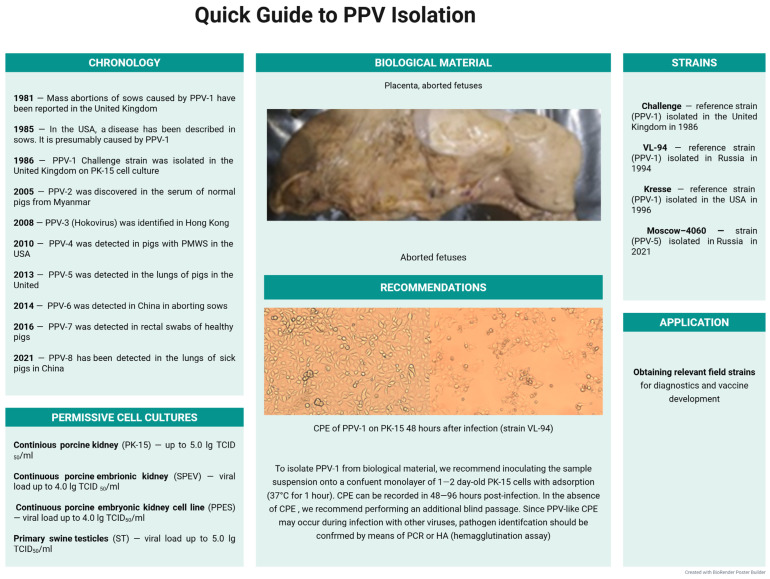
A brief history of the development of virological methods for porcine parvoviruses isolation and practical recommendations for laboratory personnel. The guide was designed using BioRender Poster Builder.

**Figure 9 microorganisms-13-02658-f009:**
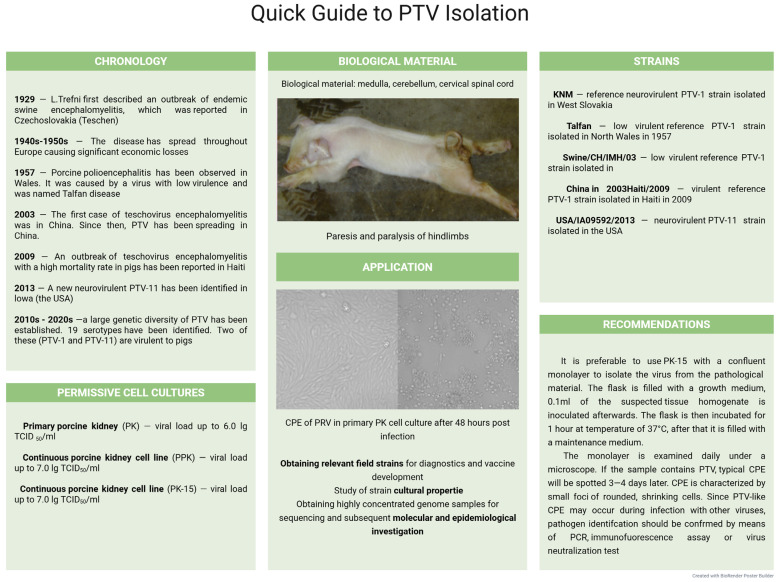
A brief history of the development of virological methods for porcine teschoviruses isolation and practical recommendations for laboratory personnel. The guide was designed using BioRender Poster Builder.

**Figure 10 microorganisms-13-02658-f010:**
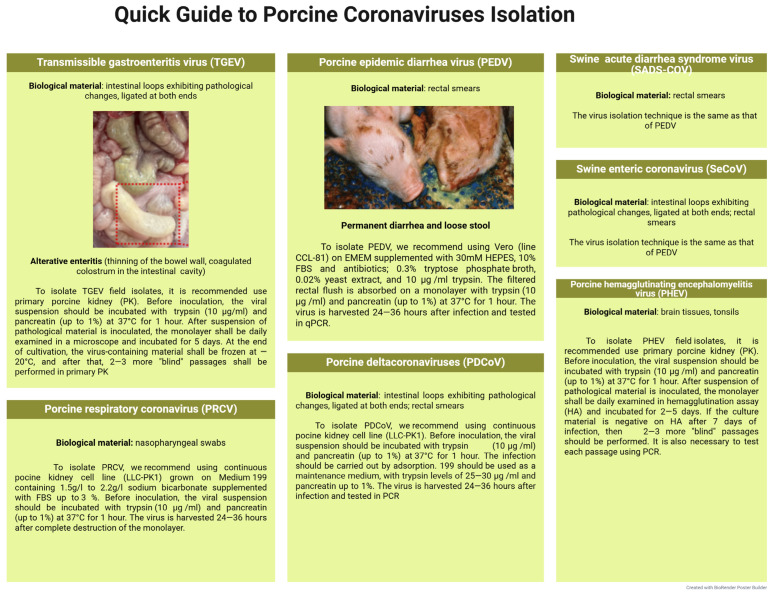
Simplified recommendations for porcine coronavirus isolation. The guide was designed using BioRender Poster Builder.

**Figure 11 microorganisms-13-02658-f011:**
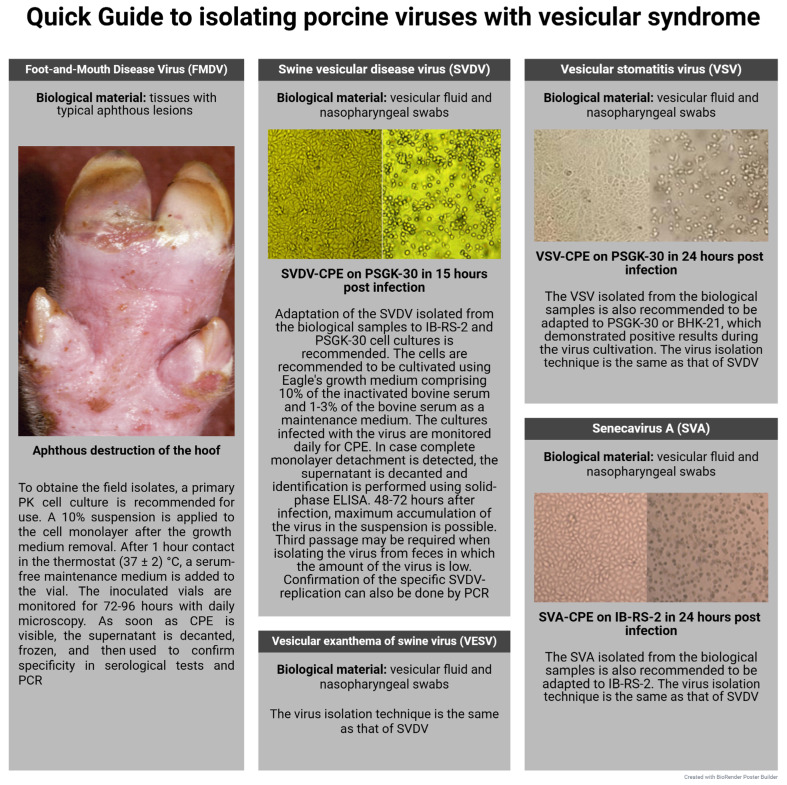
Simplified recommendations for isolating porcine viruses with vesicular syndrome. The guide was designed using BioRender Poster Builder.

**Table 1 microorganisms-13-02658-t001:** The recommended biological materials for optimal virus isolation.

No.	Viral Swine Disease	Optimal Biological Material	References
1	African swine fever	Anticoagulant (EDTA)-treated blood, spleen, lymph nodes	[16]
2	Classical swine fever	Tonsils, spleen, kidneys, Ileum, lymph nodes	[17]
3	Aujesky’s disease (pseudorabies)	Brain or trigeminal ganglion (Gasserian ganglion), lungs, tonsils, spinal cord; nasopharyngeal swabs stored in a transport medium at 4 °C for 72 h max.	[22,23]
4	Rotavirus infection	Intestinal loops exhibiting pathological changes, ligated at both ends, feces	[24]
5	Porcine respiratory and reproductive syndrome	Bronchoalveolar lavages, tonsils, lungs, lymph nodes, spleen, aborted fetuses (liver and lungs)	[18]
6	Teschen’s disease (teschovirus encephalomyelitis)	Medulla, cerebellum, cervical spinal cord	[19]
7	Swine pox	Papular–pustular eruptions on skin and mucosa (tongue, cheeks, eyelids, snout), pustule contents	[25]
8	Swine influenza	Bronchoalveolar lavages, lungs, nasopharyngeal swabs	[20]
9	Porcine parvovirus infection	Placenta, aborted fetuses (liver and lungs)	[26,27]
10	Swine enteric coronavirus disease	Intestinal loops exhibiting pathological changes, ligated at both ends, small bowel tissue smears	[28]
11	Porcine circovirus infection	In cases of post-weaning multisystemic wasting syndrome: thymus, spleen, and lymph nodes. In cases of porcine dermatitis and nephropathy syndrome: kidneys. In cases of reproductive disorders: aborted fetuses, boar semen, and dead piglet brain	[29]
12	Vesicular porcine diseases	Tissues with typical aphthous lesions (in case of foot-and-mouth disease), vesicular fluid, and nasopharyngeal swabs (in case of SVD, VES, VS)	[21]

Note: Phosphate-buffered saline (pH 7.2–7.4) or Hank’s solution can be used as a transport medium for storage and delivery of swabs, with a volume of 1–2 mL per sample.

## Data Availability

No new data were created or analyzed in this study. Data sharing is not applicable to this article.

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
