# Peer review of "Evolution of Porcine Virus Isolation: Guidelines for Practical Laboratory Application"

_microorganisms, 2025, doi:10.3390/microorganisms13122658_

Round 1

Reviewer 1 Report

Comments and Suggestions for Authors

Moiseenko and coauthors have submitted an extensive review on methods of isolating porcine viruses. On 52 pages (Including references) the "Guidelines for Laboratory Practical Application" cover isolation protocols for more than 12 relevant pig viruses  along with a description of clinical features, and history of detection. It rather reads like a book chapter and it is unusual to send it to a MDPI journal for publication. But in fact it increases the value as published books are often difficult to obtain or are overly expensive.

Isolation of viruses and propagation in cell culture was a hot topic of the 1950's to the 1970's. Looking at the references many citations precede 1980. At those days the virus characterisation was key to diagnostics and required much skill and experience. With the onset of highly sensitive immunological and molecular methods, ELISA, PCR,  qPCR, and sequencing the need for isolation and propagation became less urgent. Nevertheless, it is a good idea to put back isolation on the to do list. What euphemistically is often called "isolate" in eg. Genbank entries, often does only exist as sequence, and not as physical virus. 

I consider these "Guidelines " as a valuable contribution that comfortably revives almost fossilized publications that are not easily accessible. However, the submission's title  is  "Evolution of Virological Methods for Porcine Disease Diagnosis" and honestly, I do not understand where "evolution" becomes a topic. For the most relevant viruses the "Terrestrial manual of the OIE" is still the gold standard and like this the authors should make a stronger point in discussing the modern processes of virus identification and  the need for virus isolation. This is an issue, since many (most?) virus diagnostic labs have abandoned cell culture and virus isolation at least a decade ago. 

We learn a lot about the standard swine cell lines, PK15, SK6. MA-104, MARC 45 and others. There are many cell lines out there, PK15 from lab A is likely not the same as PK15 in lab B  Prolongated cultivation, contaminations such as M.hyo or other cells are not rare and alter the cell lines properties. Further to this we learn little about new cell lines such as ZMACs for PRRSV propagation.

specific remarks:

line 112:  "300 thousand rpm for 3 minutes".

There is no centrifuge capable of this speed.

line 243: "Most CSFV isolates and strains do not exhibit CPE during propagation in cell cultures [77-79]. This may be attributed to the RNase activity of glycoprotein Erns".

Please be careful with this statement, this was reported once and never confirmed despite several groups trying hard.

line 271: "The titer is expressed in cell-culture infectious dose (CCID) units". In line 283 it is stated “titer reached 6.9-7.3 lg TCID50/ml".

The use of CCID or TCID50 is somewhat interchangeable and not specific for CSFV. You also find the quantitation using pfu/ml. I would recommend to stick to TCID50 as this is the most prevalent term.

line 302: "Furthermore, similar to other pestiviruses, CSFV poses significant contamination risks in continuous cell line cultures due to lack of cytopathic activity when co-cultured with other viral pathogens [117-118]".  

A much greater threat - and this should be highlighted - poses the persistent infection of porcine cells with BVDV 1, -2 or Hobi like viruses. Even ATCC cell lines are frequently contaminated with BVDV. Ruminant viruses are often present in commercial fetal calf serum and cause a significant loss of susceptibility towards CSFV infection due to superinfection exclusion. Hence, all labs employing porcine cell lines should test for pestivirus infection on a regular basis. 

line 550: "The pathogen is classified into two major types (autonomous species): European (subgenus Eurpobartevirus, species Betaarterivirus suid 1 (with the prototypical strain Lelystad) and American (subgenus Ampobartevirus, species Betaarte- rivirus suid 2 (with the prototypical strain VR-2332).

According to current ICTV taxonomy, PRRSV 1 and PRRVS 2 are different species, not different types.

line 561: "Researchers have been making the first assumptions about the occurrence of RRRSV since 1912, when 14 wild boars were imported into North Carolina (USA). Some of these animals were infected with a precursor to the current RRRSV"

Please provide a reference for this claim

line 598: "The assessment of the MARC-145 cell line's permissivity to PRRSV demonstrated its promising potential for primary virus isolation. At an MOI of 0.1 TCIDâ‚…â‚€/cell, the titer ofthe Lelystad strain reached 5.04 lg TCIDâ‚…â‚€/ml by 96 hours post-inoculation (hpi), indicating high infectious activity."

For isolating PRRSV 1 field strains MA 104 or MARC 45 cells are mostly useless without tedious adaptation. Lelystad virus was already adapted to MA104 cells when the virus was studied in 1991. Same is true for PRRSV 2.

line 644: "Porcine circovirus infection is a persistent multi-syndromic disease that clinically manifests itself as secondary".

I don't understand this statement, please rephrase.

line 969: "F.Leffler and P.Frosh," should read Loeffler and Frosch

Fig.: 11, left panel. the photograph of aphtae at the pigs hoof is out of focus and should be replaced.

Author Response

Thank you for your in-depth analysis of our work. I am pleased to see that the opinion agrees with ours regarding the importance of using cell cultures to isolate porcine viruses.

We listened to you and changed the title of the manuscript. The "evolution" remained, so the chronological order of improvement of virus isolation methods is given for the main viruses. As for the guidelines for identifying each virus using PCR, IFA or other methods, they are provided in quick guides for each virus.

Comment 1; line 112:  "300 thousand rpm for 3 minutes". There is no centrifuge capable of this speed.

Response 1: You're right. This is a technical error. We fixed it.

Comment 2: line 243: "Most CSFV isolates and strains do not exhibit CPE during propagation in cell cultures [77-79]. This may be attributed to the RNase activity of glycoprotein Erns". Please be careful with this statement, this was reported once and never confirmed despite several groups trying hard.

Response 2: Yes, you're right. It was just a hypothesis. We noted this in the revised version.

Comment 3: line 271: "The titer is expressed in cell-culture infectious dose (CCID) units". In line 283 it is stated “titer reached 6.9-7.3 lg TCID50/ml". The use of CCID or TCID50 is somewhat interchangeable and not specific for CSFV. You also find the quantitation using pfu/ml. I would recommend to stick to TCID50 as this is the most prevalent term.

Response 3: Your opinion is quite popular in this regard. However, in our opinion, CCIID and TCID are not interchangeable concepts. In essence, TCID are tissue cytopathic infectious doses, and CCID are tissue non-cytopathic infectious doses. In the first case, a CPE is registered, in the second case it is not, additional Identification methods are needed. In the text about CSFV, TCID is written only in cases where the virus has manifested CPE. In other cases, CCID is used.

Comment 4: line 302: "Furthermore, similar to other pestiviruses, CSFV poses significant contamination risks in continuous cell line cultures due to lack of cytopathic activity when co-cultured with other viral pathogens [117-118]".  A much greater threat - and this should be highlighted - poses the persistent infection of porcine cells with BVDV 1, -2 or Hobi like viruses. Even ATCC cell lines are frequently contaminated with BVDV. Ruminant viruses are often present in commercial fetal calf serum and cause a significant loss of susceptibility towards CSFV infection due to superinfection exclusion. Hence, all labs employing porcine cell lines should test for pestivirus infection on a regular basis. 

Response 4: We absolutely agree with this. We noted this fact in the revised version.

Comment 5: line 550: "The pathogen is classified into two major types (autonomous species): European (subgenus Eurpobartevirus, species Betaarterivirus suid 1 (with the prototypical strain Lelystad) and American (subgenus Ampobartevirus, species Betaarte- rivirus suid 2 (with the prototypical strain VR-2332). According to current ICTV taxonomy, PRRSV 1 and PRRVS 2 are different species, not different types.

Response 5: We agree. The taxonomy has been updated.

Comment 6: line 561: "Researchers have been making the first assumptions about the occurrence of RRRSV since 1912, when 14 wild boars were imported into North Carolina (USA). Some of these animals were infected with a precursor to the current RRRSV" Please provide a reference for this claim

Response 6: The reference is provided

Comment 7: line 598: "The assessment of the MARC-145 cell line's permissivity to PRRSV demonstrated its promising potential for primary virus isolation. At an MOI of 0.1 TCIDâ‚…â‚€/cell, the titer ofthe Lelystad strain reached 5.04 lg TCIDâ‚…â‚€/ml by 96 hours post-inoculation (hpi), indicating high infectious activity." For isolating PRRSV 1 field strains MA 104 or MARC 45 cells are mostly useless without tedious adaptation. Lelystad virus was already adapted to MA104 cells when the virus was studied in 1991. Same is true for PRRSV 2.

Response 7: We agree that PAM is better suited to isolation. However, there is conflicting evidence in the literature that MARC-145 may also be suitable. In this text, we note that it is preferable to use PAM.

Comment 8: line 644: "Porcine circovirus infection is a persistent multi-syndromic disease that clinically manifests itself as secondary". I don't understand this statement, please rephrase.

Response 8: This statement was rephrased.

Comment 9: line 969: "F.Leffler and P.Frosh," should read Loeffler and Frosch

Response 9: Last names have been corrected.

Comment 10: Fig.: 11, left panel. the photograph of aphtae at the pigs hoof is out of focus and should be replaced.

Response 10: We have provided a new photo of the best quality.

Reviewer 2 Report

Comments and Suggestions for Authors

The title could be changed to "viral isolation" rather than "virological methods," since the review focuses on viral isolation.

Abstract

Lanes 7-9: Mention that the specificity of viral replication in cultures depends on the necessary cellular receptors.

Lanes 20-21: More than a promising option, it is a reality in its routine use.

Introduction

Lane 45: It is controversial whether viral isolation is a highly sensitive test. This is true only when complementary tests are used; simply observing the damage would not be sufficient.

Lane 51: It is incorrect to consider serological diagnosis as an indirect method.

Lane 61: The word "combating" is unclear in the statement.

Lane 62: Just as the limitations of this method are mentioned, the limitations of other methods for detecting viral infection should also be mentioned. This is mainly related to the samples; for example, failure to maintain the cold chain affects the possibility of successful viral isolation. Modify

Lane 88, Indicate samples used in live animals

Table 1. In the footnote, describe the recommended transport medium (e.g., number 3) and the recommended volume.

Numbers 5 and 8, bronchoalveolar lavages are a sample of choice; I recommend including them.

Number 5, include organ samples from aborted fetuses.

Number 9, of the aborted fetuses, which organ would be the preferred choice?

Lane 106, it is recommended to mention antifungals in this solution.

Lanes 94-119, mention the sample size of the recommended organs, the number of sections to be taken, how to collect and maintain them, the containers used, and the storage temperature. Supplement this section. Specify how long and at what temperature the samples should be kept and what makes them potentially useful for further work. In some cases, the samples must be cryopreserved, and their shelf life is short. Lane 119, adjust to more universal values ​​(2 atm).

ASFV subchapter: mention that several viruses produce HAD in cell culture; this is not an exclusive property of this virus (regardless of the fact that there are ASFV strains that do not have these biological properties).

Lane 275, clarify what is meant by cross-contamination in the text.

Verify the correct words and names, for example: Aujeszky's disease, lg, RRRSV, continuum.

Lanes 397-398, the same would apply in all cases, so presenting a guide for viral isolation would not be necessary; therefore, I recommend omitting this statement.

Fig. 3, mention the WOAH gold standard for ADV.

Give recommendations for incubation times for each virus when isolation is intended in cell culture.

In all figures where photographs of cell cultures with cytopathic effects are presented, mention the observed effect within the figure itself. In cases where viruses are not zoonotic, clearly state this.

Replace the word "symptoms" with "clinical signs."

When describing hemagglutination, specify the erythrocyte species used for the test.

Lanes 409-412: Clarify whether these cases in different species are due to specific RV or rotaviruses.

Lane 520: Specify the type and concentration of trypsin used.

Mention that, in addition to the cytopathic effect in cultures or chicken embryos, supernatants or allantoic fluid can be collected and tested for hemagglutination.

Fig. 5: Include the H1N1 pandemic influenza event (2009) in both the text and the figure. This figure is not in the same order as the previous figures.

Lanes 549-554: Update the viral taxonomy in all cases, as the names have been recently modified. Figure 7, clearly indicate that permissive cell cultures are for the isolation of PCV-2.

Lanes 875-876, mention the background in the USA, which was the first country in the Americas to receive the infection on the continent.

Figure 11, improve the image of FMDV lesions.

For other viruses, I recommend mentioning Orthorubulavirus suis and Niphavirus.

Author Response

Thank you for your detailed analysis of our manuscript. Most of your comments have been taken into account and we have corrected them. We have also considered your suggestion for a different title for the article.

Comment 1: Lanes 7-9: Mention that the specificity of viral replication in cultures depends on the necessary cellular receptors.

Response 1: We agree. We clarified that we mean the cytopathic effect.

Comment 2: Lanes 20-21: More than a promising option, it is a reality in its routine use.

Response 2: We agree. The wording has been clarified.

Comment 3: Lane 45: It is controversial whether viral isolation is a highly sensitive test. This is true only when complementary tests are used; simply observing the damage would not be sufficient.

Response 3: We agree. This fact has been added to the text.

Comment 4: Lane 51: It is incorrect to consider serological diagnosis as an indirect method.

Response 4: Serological methods for detecting antibodies, rather than the antigens of a virus, are indirect, as they do not detect viruses (their genome or antigens), but aim to detect antibodies to causative agent of disease. If the serological reaction reveals the virus antigen, then it is certainly direct.

Comment 5: Lane 61: The word "combating" is unclear in the statement.

Response 5: We agree. The wording has been clarified.

Comment 6: Lane 62: Just as the limitations of this method are mentioned, the limitations of other methods for detecting viral infection should also be mentioned. This is mainly related to the samples; for example, failure to maintain the cold chain affects the possibility of successful viral isolation. Modify

Response 6: We agree. This fact has been added to the text.

Comment 7: Lane 88, Indicate samples used in live animals

Response 7: This fact has been added to the text.

Comment 8: Table 1. In the footnote, describe the recommended transport medium (e.g., number 3) and the recommended volume.

Response 8: The composition of the transport environment was added in a footnote.

Comment 9: Numbers 5 and 8, bronchoalveolar lavages are a sample of choice; I recommend including them.

Response 9: We agree. This fact has been added to the text.

Comment 10: Number 5, include organ samples from aborted fetuses. Number 9, of the aborted fetuses, which organ would be the preferred choice?

Response 10: This fact has been added to the text.

Comment 11: Lane 106, it is recommended to mention antifungals in this solution.

Response 11: The minimum size of fungi conidia is 2 μm, and the filter size is 450 nanometers. Adding antimycotic agents to the filtrate is largely useless.

Comment 12: Lanes 94-119, mention the sample size of the recommended organs, the number of sections to be taken, how to collect and maintain them, the containers used, and the storage temperature. Supplement this section. Specify how long and at what temperature the samples should be kept and what makes them potentially useful for further work. In some cases, the samples must be cryopreserved, and their shelf life is short. Lane 119, adjust to more universal values ​​(2 atm).

Response 12: In general, information has been added to the text. As for the amount of material, it is all regulated by local regulations.

Comment 13: ASFV subchapter: mention that several viruses produce HAD in cell culture; this is not an exclusive property of this virus (regardless of the fact that there are ASFV strains that do not have these biological properties).

Response 13: You're right. Information about other hemadsorbing viruses has been added to the text.

Comment 14: Lane 275, clarify what is meant by cross-contamination in the text.

Response 14: We agree. The wording has been clarified.

Comment 15: Verify the correct words and names, for example: Aujeszky's disease, lg, RRRSV, continuum.

Response 15: The wording has been clarified.

Comment 16: Lanes 397-398, the same would apply in all cases, so presenting a guide for viral isolation would not be necessary; therefore, I recommend omitting this statement.

Response 16: We agree. The sentense was removed.

Comment 17: Fig. 3, mention the WOAH gold standard for ADV.

Response 17:  According to the WOAH Manual, the PRV isolation is considered one of the diagnosis methods. However, name it the gold standard, as in the case of ASFV and CSFV, is too strong, since PCR is generally preferred.

Comment 18: Give recommendations for incubation times for each virus when isolation is intended in cell culture.

Response 18: The original text of the manuscript shows the time of observation, manifestations of CPE, and cultivation of viruses (in a quick guide).

Comment 19: In all figures where photographs of cell cultures with cytopathic effects are presented, mention the observed effect within the figure itself. In cases where viruses are not zoonotic, clearly state this.

Response 19: In most cases, CPE manifests in the same way (through desadhesion and cell destruction). To be honest, we don't see the need to overload quick guides with information. Regarding ASFV, CSFV and porcine circoviruses information is given in the notes to the picture.

Comment 20: Replace the word "symptoms" with "clinical signs."

Response 20: The wording has been clarified.

Comment 21: Lanes 409-412: Clarify whether these cases in different species are due to specific RV or rotaviruses.

Response 21: The information has been updated. These cases were reported in humans and all were caused by rotaviruses (but not necessarily porcine).

Comment 22: Lane 520: Specify the type and concentration of trypsin used.

Response 22: The information has been added.

Comment 23: Mention that, in addition to the cytopathic effect in cultures or chicken embryos, supernatants or allantoic fluid can be collected and tested for hemagglutination.

Response 23: The details can be found in the quick guide for swine influenza virus

Comment 24: Fig. 5: Include the H1N1 pandemic influenza event (2009) in both the text and the figure. This figure is not in the same order as the previous figures.

Response 24: The pandemic caused by the H1N1 virus in 2009 is unrelated to swine influenza. This virus is a triple reassortant that originated in pigs and passed on to humans. This virus has never circulated among pigs.

Comment 25: Lanes 549-554: Update the viral taxonomy in all cases, as the names have been recently modified. Figure 7, clearly indicate that permissive cell cultures are for the isolation of PCV-2.

Response 25: The taxonomy has been clarified in all cases.

Comment 26: Lanes 875-876, mention the background in the USA, which was the first country in the Americas to receive the infection on the continent.

Response 26: We agree. This fact has been added.

Comment 27: Figure 11, improve the image of FMDV lesions.

Response 27: A new photo with the best quality has been provided.

Comment 28: For other viruses, I recommend mentioning Orthorubulavirus suis and Niphavirus.

Response 28: We totally agree. Information about these two viruses has been added.

Round 2

Reviewer 2 Report

Comments and Suggestions for Authors

I thank the authors for addressing the comments; they have managed to resolve most of them.